# Respiratory syncytial virus co-opts host mitochondrial function to favour infectious virus production

**MengJie Hu[1,2], Keith E Schulze[3], Reena Ghildyal[4], Darren C Henstridge[5], Jacek L Kolanowski[6‡], Elizabeth J New[6], Yuning Hong[7], Alan C Hsu[8], Philip M Hansbro[8], Peter AB Wark[8], Marie A Bogoyevitch[1†], David A Jans[2†]***

[1]Department of Biochemistry and Molecular Biology, University of Melbourne, Melbourne, Australia; [2]Department of Biochemistry and Molecular Biology, Monash University, Melbourne, Australia; [3]Monash Micro Imaging, Monash University, Melbourne, Australia; [4]Centre for Research in Therapeutic Solutions, Faculty of Science and Technology, University of Canberra, Canberra, Australia; [5]Baker Heart and Diabetes Institute, Melbourne, Australia; [6]School of Chemistry, The University of Sydney, Sydney, Australia; [7]Department of Chemistry and Physics, La Trobe Institute for Molecular Science, La Trobe University, Melbourne, Australia; [8]Priority Research Centre for Healthy Lungs, Hunter Medical Research Institute (HMRI) and School of Biomedical Sciences and Pharmacy, University of Newcastle, Newcastle, Australia

**\*For correspondence:**
David.Jans@monash.edu

†These authors contributed equally to this work

**Present address:** ‡Institute of Bio-organic Chemistry, Polish Academy of Sciences, Poznan, Poland

**Competing interests:** The authors declare that no competing interests exist.

**Abstract** Although respiratory syncytial virus (RSV) is responsible for more human deaths each year than influenza, its pathogenic mechanisms are poorly understood. Here high-resolution quantitative imaging, bioenergetics measurements and mitochondrial membrane potential- and redox-sensitive dyes are used to define RSV's impact on host mitochondria for the first time, delineating RSV-induced microtubule/dynein-dependent mitochondrial perinuclear clustering, and translocation towards the microtubule-organizing centre. These changes are concomitant with impaired mitochondrial respiration, loss of mitochondrial membrane potential and increased production of mitochondrial reactive oxygen species (ROS). Strikingly, agents that target microtubule integrity the dynein motor protein, or inhibit mitochondrial ROS production strongly suppresses RSV virus production, including in a mouse model with concomitantly reduced virus-induced lung inflammation. The results establish RSV's unique ability to co-opt host cell mitochondria to facilitate viral infection, revealing the RSV-mitochondrial interface for the first time as a viable target for therapeutic intervention.
DOI: https://doi.org/10.7554/eLife.42448.001

## Introduction

Respiratory syncytial virus (RSV), an enveloped RNA virus of the *Pneumoviridae* family, is a leading cause of serious lower respiratory tract illness in infants and a potent respiratory pathogen in elderly and immunosuppressed adults (*Nair et al., 2010*; *Hall et al., 2009*), leading to more deaths each year worldwide than influenza. Despite this, there are no effective anti-RSV therapeutics generally available, with palivizumab (Synagis) and ribavirin the only approved agents as a prophylactic and therapeutic, respectively, for high-risk patients (*Hurwitz, 2011*; *Hebert and Guglielmo, 1990*; *Resch, 2017*). Like all pneumoviruses, RSV replicates in the cytoplasm (*Collins et al., 2013*), but specific interaction with host cell organelles, and the mitochondria in particular, has remained largely

unexplored. Unbiased discovery studies capitalising on quantitative proteomic protocols to identify changes in protein levels upon RSV infection have revealed a significant impact on the abundance of a number of nuclear-encoded mitochondrial proteins (*Munday et al., 2015*; *van Diepen et al., 2010*; *Kipper et al., 2015*), including respiratory complex I proteins, outer mitochondrial membrane complex subunits, voltage-dependent anion channel protein, and the prohibitin subunits that play essential roles in the regulation of mitochondrial dynamics, morphology and biogenesis (*Kipper et al., 2015*). The implication is that RSV may have the capacity to impact host cell mitochondrial activities, and in keeping with this, we recently were able to document changes in mitochondrial morphology during RSV infection (*Hu et al., 2017*).

Mitochondria are integral to ATP production and reactive oxygen species (ROS) metabolism in eukaryotic cells. Oxidative phosphorylation driven by ATP synthase/complex V and the electron transport chain (complexes I-IV) is responsible for up to 90% of cellular ATP production (*Schertl and Braun, 2014*; *Letts et al., 2016*). The electron transport chain carries out a series of redox reactions, which are tightly coupled to the generation of mitochondrial membrane potential ($\Delta\psi_m$) through proton translocation across the inner mitochondrial membrane to drive ATP synthesis (*Schertl and Braun, 2014*; *Letts et al., 2016*). ROS arising from incomplete electron transfer across complexes I and III are generally cleared by intracellular antioxidant enzymes under normal conditions (*Schertl and Braun, 2014*; *Letts et al., 2016*), but oxidative stress can occur when ROS production exceeds antioxidant capacity (*Lin and Beal, 2006*; *Schieber and Chandel, 2014*). Changes in cytoskeletal organization and/or motor activities can impact mitochondrial distribution and function because mitochondria are trafficked intracellularly through the action of molecular motors operating on microtubules and actin filaments (*Welte, 2004*; *Hancock, 2014*).

Here the RSV-host interface at the level of mitochondrial organization and function is interrogated in detail for the first time. A unique combination of redox/membrane potential-sensitive/ratiometric dyes, direct bioenergetics analyses, and high-resolution quantitative imaging/flow cytometric analysis is used to demonstrate that RSV drives a staged redistribution of mitochondria in microtubule- and dynein-dependent fashion, concomitant with compromised mitochondrial respiration in infected cells. Inhibiting RSV-induced changes in mitochondrial distribution both restores mitochondrial respiration, and can protect against RSV infection. Further, we show that RSV's effects on the mitochondria result in enhanced mitochondrial ROS production; importantly, blocking mitochondrial ROS with a specific inhibitor significantly reduces RSV replication and titers, and alleviates RSV-induced inflammation in a mouse model. The results highlight RSV's ability to co-opt the host cell mitochondria to enhance mitochondrial ROS to facilitate virus production, and establish it for the first time as a viable target for future anti-RSV strategies.

## Results

### RSV infection drives mitochondrial perinuclear clustering and redistribution of mitochondria towards the microtubule organizing centre (MTOC)

Building on our preliminary observations of altered mitochondrial morphology in RSV-infected cells (*Hu et al., 2017*), we first performed high resolution Airyscan CLSM imaging of mitochondria in RSV-infected cells at 8 hr post-infection (p.i.) (*Figure 1A*). Mock- and RSV-infected cells exhibited fragmented, tubular and fibrillar morphologies as revealed by MitoTrackerRed staining (*Figure 1A*), with a higher percentage of infected cells showing fragmented morphology, and a lower percentage with tubular and fibrillar morphologies, compared to uninfected cells (*Hu et al., 2013*). Quantitative analysis confirmed the observations, with a significant ($p < 0.001$)>30% increase of cells showing fragmented mitochondrial morphology following infection, compared to mock infected cells (*Figure 1B*).

Clear differences in mitochondrial distribution between mock- and RSV-infected cells were also evident, whereby perinuclear clusters of mitochondria could be clearly observed in the infected cells in stark contrast to the even distribution of mitochondria throughout mock-infected cells (*Figure 1A*); dynamic perinuclear clustering in the infected but not non-infected cells could be visualised by live cell imaging (compare *Figure 1—videos 1* and *2*, where mitochondria are imaged using the CellLight Mitochondria-RFP BacMam 2.0* system). Quantitative analysis to determine the $R_{90\%}$ parameter, the radius of a circle required to enclose 90% of the MitoTrackerRed fluorescence

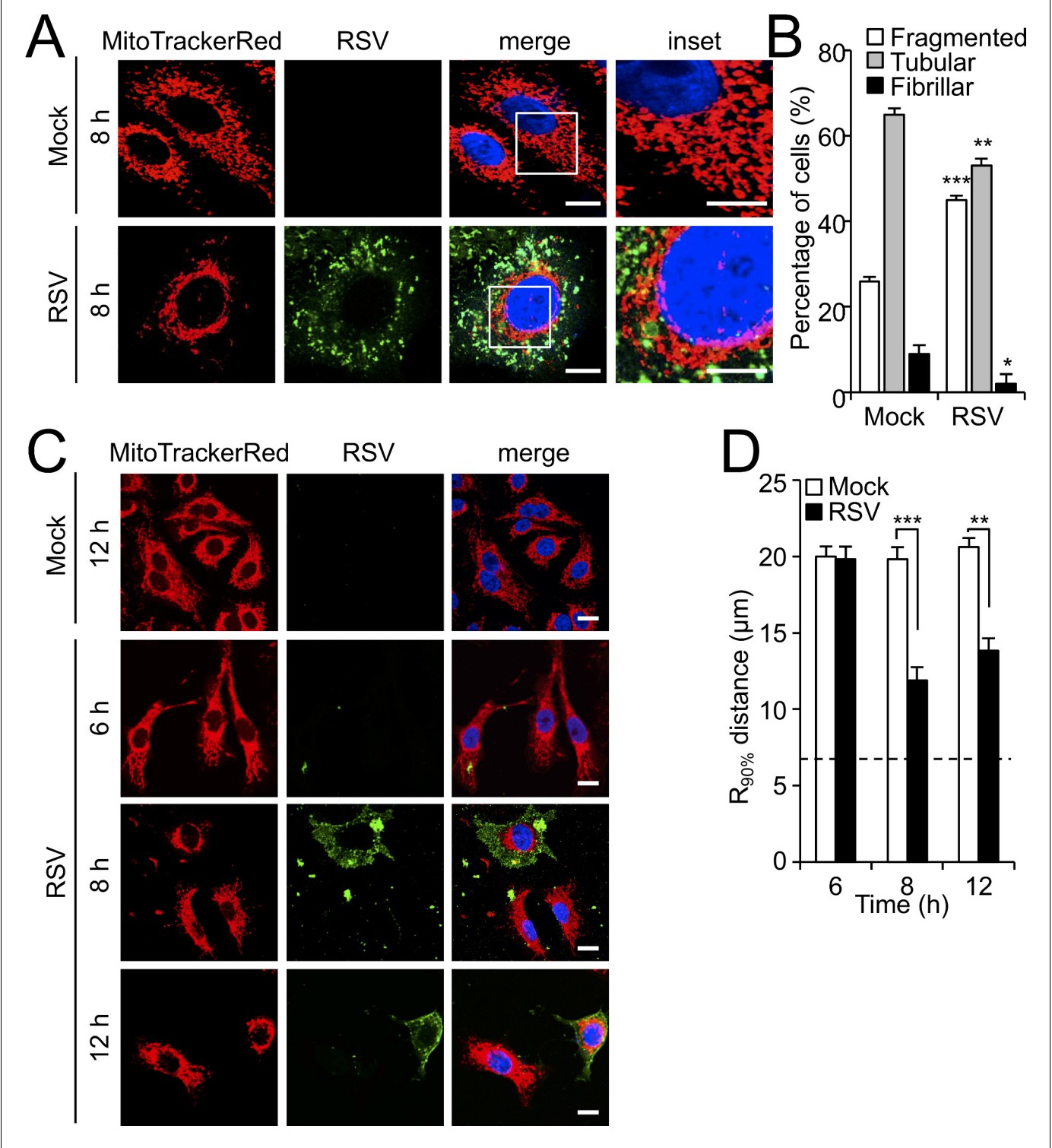

**Figure 1.** RSV infection induces mitochondrial perinuclear clustering early in infection. A549 cells were mock- or RSV-infected (MOI 1) for the times indicated, followed by staining for mitochondria (MitoTrackerRed), RSV infection (RSV antibody, green) and nuclei (DAPI, blue). (**A**) Cells were imaged by Airyscan super-resolution CLSM. Merge panels overlay all three stains; inset (right panels) corresponds to the boxed regions. Scale bar = 5 μm. (**B**) Quantification of mitochondrial morphologies following RSV infection. Cells with predominantly fragmented, tubular or fibrillar mitochondrial morphologies, defined by width/length ratios of 1:1, 1:3 and 1:10 respectively (*Hu et al., 2013*) were scored by assessing 25–30 cells per condition on three independent occasions. Data represent the mean ± SEM; ***$p<0.001$, **$p<0.01$, *$p<0.05$ relative to the mock. (**C and D**) Cells were imaged by CLSM. (**C**) Merge panels overlay all three stains. Scale bar = 10 μm. (**D**) Perinuclear radial distribution of mitochondria ($R_{90\%}$; see Materials and methods)

*Figure 1 continued on next page*

*Figure 1 continued*

was calculated from images such as those in C). Results represent the mean ± SEM for n = 3 independent experiments, each of which analysed 25–30 cells per sample; ***p<0.001, **p<0.01. The dashed line represents the average nuclear radius.

DOI: https://doi.org/10.7554/eLife.42448.002

The following video and figure supplements are available for figure 1:

**Figure supplement 1.** Lack of impact of RSV infection on the Golgi apparatus.

DOI: https://doi.org/10.7554/eLife.42448.003

**Figure supplement 2.** RSV infection induces mitochondrial perinuclear clustering in immortalised human airway progenitor-like basal cells.

DOI: https://doi.org/10.7554/eLife.42448.004

**Figure supplement 3.** RSV-induced mitochondrial perinuclear clustering is independent of actin filaments.

DOI: https://doi.org/10.7554/eLife.42448.005

**Figure 1—video 1.** Mitochondrial dynamics in mock-infected cells.

DOI: https://doi.org/10.7554/eLife.42448.006

**Figure 1—video 2.** RSV infection induces perinuclear mitochondrial clustering.

DOI: https://doi.org/10.7554/eLife.42448.007

**Figure 1—video 3.** Nocodazole blocks perinuclear mitochondrial clustering induced by RSV infection.

DOI: https://doi.org/10.7554/eLife.42448.008

relative to the centre of the nucleus (*van Bergeijk et al., 2015*), confirmed mitochondrial perinuclear clustering at 8 and 12 h p.i. (*Figure 1C*). $R_{90\%}$ was significantly (p<0.001) reduced (over 30%) at 8 or 12 h p.i. (*Figure 1D*) compared to mock-infected cells, confirming RSV-induced perinuclear mitochondrial clustering early in infection. Intriguingly, asymmetric mitochondrial distribution was observed at later time points in infection (18 and 24 h p.i.; *Figure 2A*). By specifically staining for the microtubule organizing centre (MTOC) using an antibody against the MTOC component γ-tubulin (highlighted by arrows in the merge panels), we could show that the majority of mitochondria are situated close to the MTOC at 18 or 24 h p.i. (*Figure 2A*). This asymmetric distribution was confirmed by our quantitative analysis of the angular distributions of mitochondrial fluorescence (*Figure 2B*) towards the axis of the MTOC (red line); quantitation of mitochondrial staining within 45° on either side of the MTOC revealed significantly (p<0.001) increased (40–50%) levels in infected cells at 18 and 24 h p.i. compared to the mock-infected controls (*Figure 1C*). To confirm that the effects on host mitochondria are specific, we also tested for changes in the Golgi apparatus by staining using the CellLight Golgi-GFP *BacMam 2.0* (*Figure 1—figure supplement 1*). The Golgi cisterna remained asymmetrically close to the nuclei in mock- and RSV-infected cells (*Figure 1—figure supplement 1*), in stark contrast to the striking changes in mitochondria at 8 and 24 h p.i. Together, these findings indicate that RSV infection specifically reorganizes the host cell mitochondria, with perinuclear clustering followed by MTOC-oriented asymmetry.

To confirm these results to clinically relevant experimental systems, we explored the impact of RSV on human airway basal cells using an immortalised human airway basal cell line (BCi) derived from a healthy non-smoker (NS1) capable of multipotent differentiation and responding to extremal stimuli (*Walters et al., 2013*). BCi-NS1 cells were infected with RSV for 18 or 36 h p.i. (*Figure 1—figure supplement 2A*). In contrast to the even distribution of mitochondria throughout the mock-infected cells (*Figure 1—figure supplement 2A*; first two rows), substantial perinuclear mitochondrial clustering was observed in RSV-infected cells at 18 and 36 hr (*Figure 1—figure supplement 2A*; 3rd and 4th rows), completely consistent with our observations in A549 cells (*Figure 1*). Quantitative analysis of the $R_{90\%}$ parameter (*Figure 1—figure supplement 2B*) confirmed these observations, whereby $R_{90\%}$ was significantly (p<0.01) reduced (over 30%) in RSV-infected cells at 18 or 36 h p.i., compared to mock-infected cells. These results were consistent with the idea that RSV induces perinuclear mitochondrial clustering in a clinically relevant model of human infection.

## RSV-induced mitochondrial redistribution is microtubule- and dynein-dependent

Mitochondrial distribution is known to be controlled by cytoskeletal-associated motor proteins (*Boldogh and Pon, 2007*). To test whether RSV-induced mitochondrial redistribution requires an intact cytoskeleton, we treated infected cells with agents that depolymerize/destabilize the actin or microtubule networks: cytochalasin D (*Figure 1—figure supplement 3A*) or nocodazole

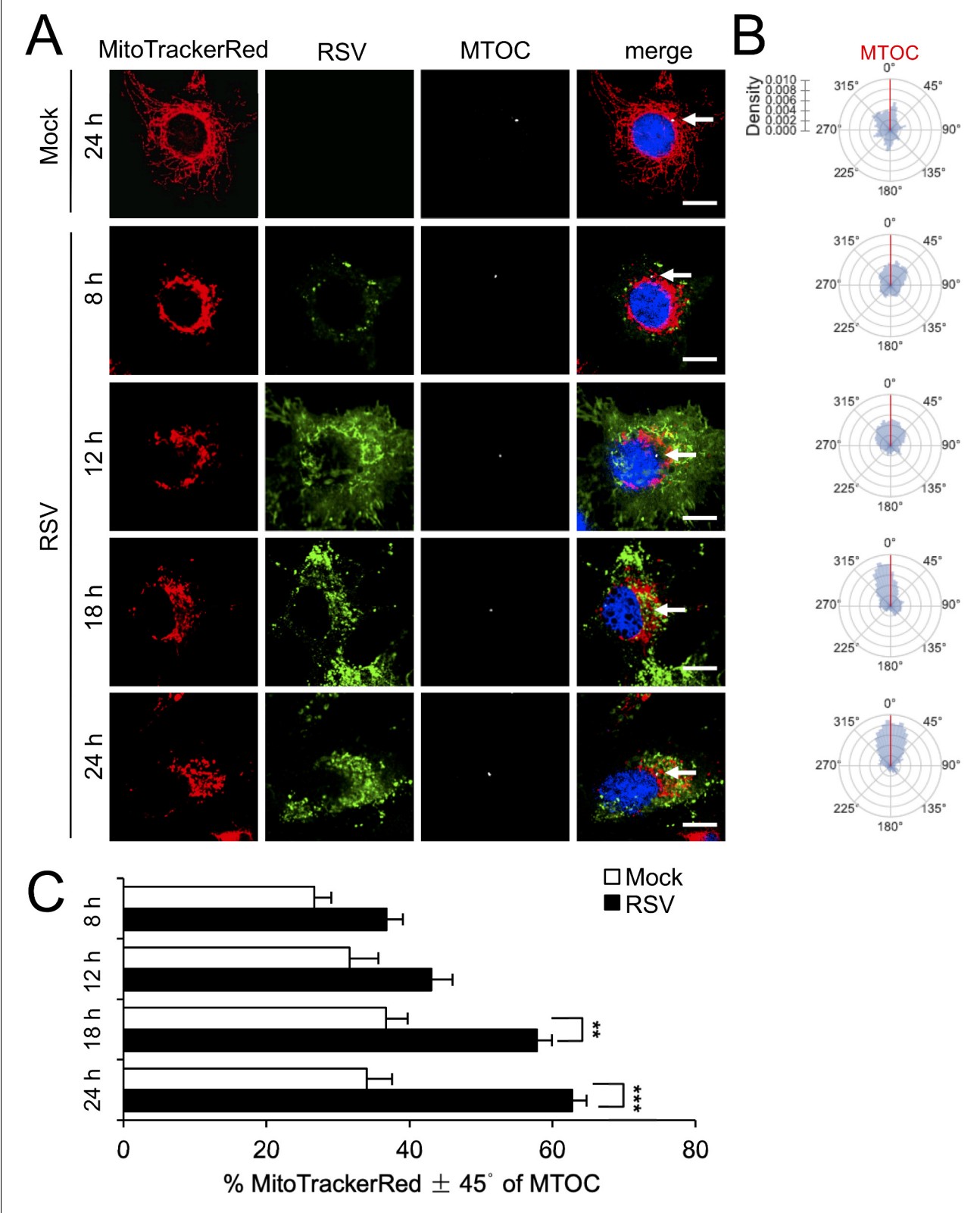

**Figure 2.** RSV infection induces asymmetric distribution of mitochondria close to the MTOC later in infection. A549 cells were mock- or RSV-infected (MOI 1) for 8–24 hr, as indicated, and then stained for mitochondria (MitoTrackerRed), RSV infection (RSV antibody, green), MTOC (γ-tubulin antibody, white) and nuclei (DAPI, blue). Cells were imaged by CLSM. (A) Merge panels overlay all four stains; arrows indicate MTOC. In all panels, scale bar = 5 μm. (B and C) Mitochondrial distribution relative to γ-tubulin staining was estimated from images such as those in A. (B) Polar kernel density plot

*Figure 2 continued on next page*

*Figure 2 continued*

showing the frequency of angles of mitochondria-stained pixels from the centre of the nucleus, normalised to the position of the MTOC (red line). (**C**) Proportion of mitochondrial signal detected within 45˚ either side of the MTOC was quantitated. Results represent the mean ± SEM for n = 3 independent experiments, each of which analysed 25–30 cells per sample; **p<0.01, ***p<0.001.
DOI: https://doi.org/10.7554/eLife.42448.009

(*Figure 3A*), respectively. Significantly, RSV-infected cells continued to show perinuclear clustering of mitochondria following treatment with cytochalasin D (*Figure 1—figure supplement 3B and C*) but not with nocodazole (*Figure 3A*, *Figure 3 - Figure 1—video 3*), confirming that RSV-induced mitochondrial perinuclear clustering is strongly dependent on the integrity of the microtubule network but not the actin cytoskeleton. Quantitative analysis for the $R_{90\%}$ parameter reinforced this observation that RSV-induced mitochondrial redistribution is microtubule-dependent (*Figure 3B*).

To test the potential role of microtubule motor proteins, we used the agents EHNA (erythro-9-[2-hydroxy-3-nonyl]adenine) and monastrol that specifically inhibit dynein-dependent (retrograde) and kinesin-dependent (anterograde) transport along microtubules, respectively. Neither treatment, in stark contrast to treatment with nocodazole, impacted the filamentous microtubule network, as indicated by α-tubulin staining (*Figure 3A*; top four rows), as expected. Strikingly, RSV-induced mitochondrial perinuclear clustering was completely abolished by treatment with EHNA but not monastrol. Quantitative analysis for the $R_{90\%}$ parameter confirmed this finding (*Figure 3B*), indicating that RSV-induced mitochondrial perinuclear clustering relies on dynein-dependent retrograde transport along intact microtubules.

To further reinforce the contribution of dynein to RSV-induced mitochondrial perinuclear clustering, we pretreated cells with small interference RNAs (siRNAs) targeting cytoplasmic dynein (*DYNLT1* or *DYNC1H1*), as well as controls of siRNA targeting kinesin light chain 1 (*KLC1*) (*Hirokawa et al., 2009*) or scrambled siRNA (scr), prior to virus infection. These treatments resulted in substantial reductions in the cognate target protein levels (*Figure 3C*), with no impact on the filamentous microtubule network, as expected (see α-tubulin staining in *Figure 3D*; top four rows). Consistent with the effects for EHNA above, RSV-induced perinuclear mitochondrial clustering was suppressed by depletion of either *DYNLT1* or *DYNC1H1*, resulting in an even distribution of reticular mitochondria (*Figure 3D*). In contrast, RSV-induced perinuclear mitochondrial clustering was not affected by depletion of *KLC1* (*Figure 3D*). Quantitative analysis for the $R_{90\%}$ parameter supported these conclusions (*Figure 3E*), confirming RSV-induced perinuclear redistribution of mitochondria to be dynein-dependent.

We extended our analysis to the RSV-induced mitochondrial asymmetry that follows mitochondrial perinuclear clustering, finding that nocodazole and EHNA treatments prevent the MTOC-oriented mitochondrial asymmetry characteristic of longer (18 hr) RSV infection (*Figure 4A*). Quantitative analysis for the angular mitochondrial distribution (*Figure 4B–4C*) supported this conclusion, results overall demonstrating a dynein/microtubule-dependent mechanism underlying RSV-induced mitochondrial redistribution during the course of infection.

## RSV infection inhibits host mitochondrial respiration dependent on dynein/microtubule integrity

The striking effects of RSV infection on mitochondrial organization prompted us to evaluate the impact of RSV infection on mitochondrial respiratory function. We used the Seahorse XF96 Extracellular Flux Analyser to measure oxygen consumption rate (OCR) and extracellular acidification rate (ECAR) of living cells over the time course (6–24 hr) of RSV infection (*Figure 5A*), as indicators of mitochondrial respiration and glycolysis, respectively (*Wu et al., 2007*). OCR progressively decreased during the RSV infection (*Figure 5A*, main panel), and was accompanied by increases in ECAR (*Figure 5A*, inset) indicating an inhibition of mitochondrial respiration and a parallel increase in glycolytic metabolism for energy production. These effects paralleled the robustness of infection, with increasing multiplicity of infection (MOI) resulting in more severe effects (*Figure 5—figure supplement 1A*).

These observations were extended by performing successive OCR measurements in the presence of oligomycin (ATP synthase inhibitor), FCCP (proton ionophore), antimycin A (mitochondrial

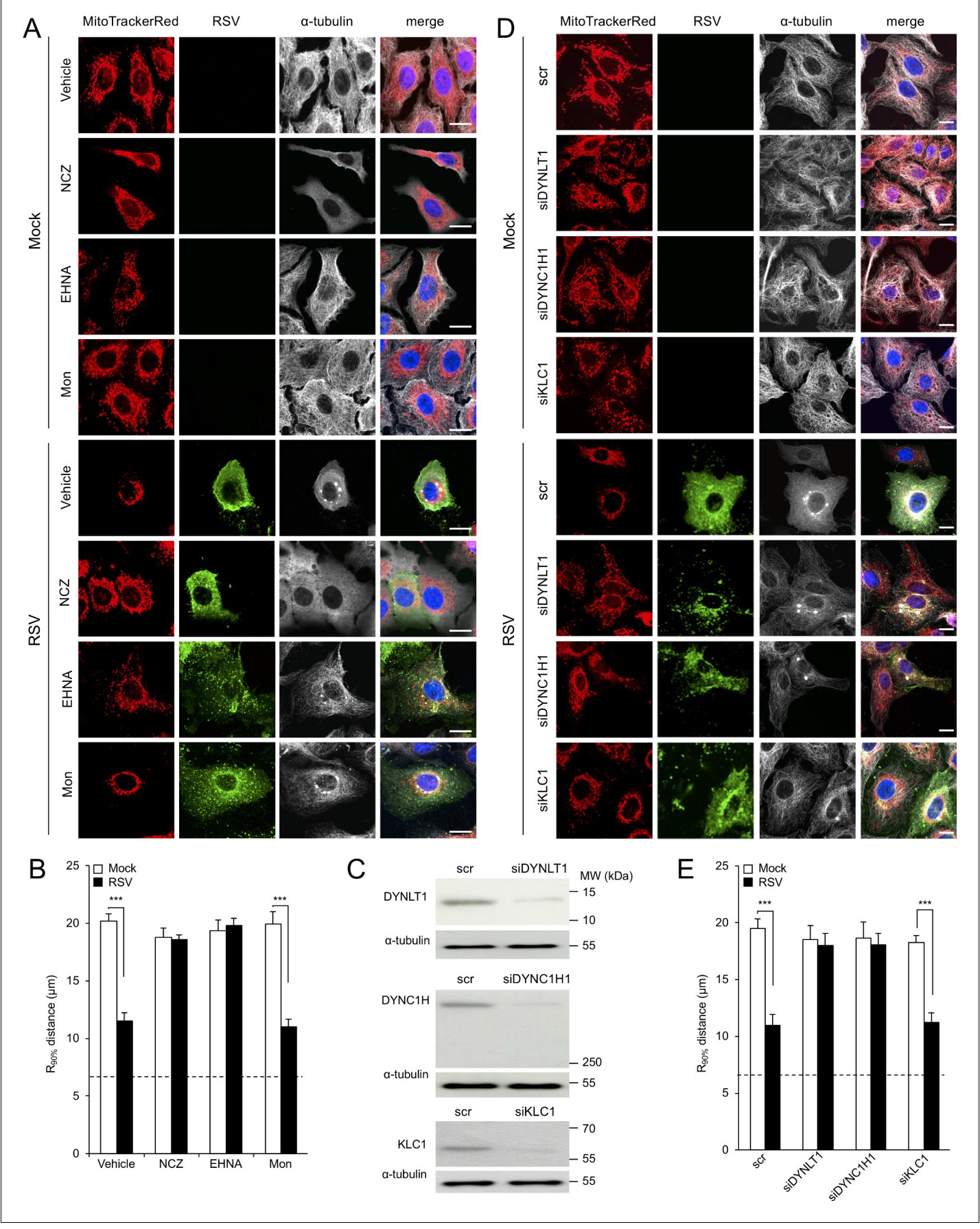

**Figure 3.** RSV-induced mitochondrial perinuclear clustering is microtubule- and dynein-dependent. A549 cells were mock- or RSV-infected (MOI 3) for 8 hr with the indicated agents added in the last 2 hr: (**A and B**) the microtubule-depolymerizing agent nocodazole (NCZ, 17 µM), the dynein inhibitor EHNA (200 µM), the kinesin inhibitor monastrol (Mon, 50 µM), or DMSO as a vehicle control. Cells were then stained for mitochondria (MitoTrackerRed), RSV (RSV antibody, green), α-tubulin (white) and nuclei (DAPI, blue), and cells imaged by CLSM. (**A**) Merge panels overlay all four stains. (**B**) The perinuclear radial distribution of mitochondria ($R_{90\%}$) was calculated as per *Figure 1D*, ***$p<0.001$. (**C–E**) A549 cells were pretreated (48 hr) with siRNA (50 nM) for dynein light chain Tctex-type 1 (*DYNLT1*), dynein cytoplasmic 1 heavy chain (*DYNC1H1*), kinesin light chain 1 (*KLC1*), or scrambled control (scr). (**C**) Immunoblot analysis for *DYNLT1*, *DYNC1H1*, *KLC1*, or the control α-tubulin, as indicated (40 µg cell lysate protein/lane). (**D and E**) RSV infection, immunostaining, and $R_{90\%}$ analysis were as per (**A and B**). In all panels, scale bar = 10 µm.

DOI: https://doi.org/10.7554/eLife.42448.010

complex III inhibitor) and rotenone (mitochondrial complex I inhibitor) (*Figure 5A*, *Figure 5—figure supplement 1A*); these are all routinely used inhibitors of specific component of the ETC, enabling the key parameters of mitochondrial metabolic activity (basal, ATP-linked, maximal and non-mitochondrial respiration activities) to be determined (*Figure 5—figure supplement 1B*). Whilst no significant changes were observed within 6 h p.i., we observed significant decreases in maximal OCR (from 8 h p.i.), basal and ATP-linked OCR (from 18 h p.i.); non-mitochondrial OCR was significantly increased from 8 h p.i. (*Figure 5B*). These effects again paralleled the robustness of infection, with increasing multiplicity of infection (MOI) resulting in more severe effects (*Figure 5—figure supplement 1C*). Thus, the impact of RSV on mitochondrial respiration was to progressively reduce ATP-linked and maximal respiratory function over the time course of infection.

To assess the extent to which RSV's impact on mitochondrial respiratory function is linked to RSV-induced microtubule-dependent mitochondrial redistribution, we treated mock- or RSV-infected cells with nocodazole or EHNA for 2 hr and then performed Seahorse OCR and ECAR analyses at 18 h p.i. (*Figure 5C*). Strikingly, all respiratory activities (basal, ATP-linked, maximal and non-mitochondrial) in the RSV-infected cells treated with nocodazole or EHNA remained unchanged (*Figure 5C*). Together, these results show that the RSV-induced changes in host cell respiration activities are dependent on dynein/microtubules.

### RSV infection decreases mitochondrial membrane potential ($\Delta\chi_m$) but enhances mitochondrial reactive oxygen species (ROS) generation to favour virus production

Mitochondrial respiration is required to maintain mitochondrial membrane potential $\Delta\psi_m$ (*Gottlieb et al., 2003*; *Dey and Moraes, 2000*). To assess how RSV impacts $\Delta\psi_m$, we infected cells with eGFP-rRSV that has been engineered to express GFP upon host cell infection (*Webster Marketon et al., 2014*) to ensure unambiguous identification of RSV-infected cells and then stained these cells with the $\Delta\psi_m$-sensitive dye tetramethylrhodamine ethyl ester (TMRE) (*Dejonghe et al., 2016*) for live cell imaging over 6–24 h p.i. (*Figure 6*). The proton ionophore FCCP was used as a control to give maximal dissipation of the $\Delta\psi_m$ as indicated by the uniform loss of TMRE fluorescence (*Figure 6A*, 2nd row of panels). Lower TMRE fluorescence was observed in RSV-infected cells at 18 and 24 h p.i., an impact strikingly apparent when infected cells were imaged alongside non-infected cells in the same field (*Figure 6A*, 5th and 6th rows of panels). Quantification of the integrated density of TMRE fluorescence confirmed the results, revealing significantly reduced ($p<0.001$) $\Delta\psi_m$ at 18 h p.i. (*Figure 6B*). To monitor these RSV-induced changes in $\Delta\psi_m$ over the period of 16–18 h p.i. in real time, we used the photobleach-resistant $\Delta\psi_m$-sensitive dye tetraphenyl-ethylene-phenyl-indolium salt (TPE-Ph-In) (*Zhao et al., 2015*), documenting that TPE-Ph-In fluorescence was maintained throughout the 2 hr imaging period in mock-infected cells, but showed a progressive loss of TPE-Ph-In fluorescence at 16 to 18 h p.i. in RSV-infected cells (*Figure 6—figure supplement 1*). Thus, by ~18 h p.i., RSV infection induces a loss of host cell mitochondrial membrane potential, $\Delta\psi_m$.

In addition to being critical contributors to ATP generation, mitochondria are important sites of reactive oxygen species production, with modulation of mitochondrial intracellular location recently gaining interest as a critical mechanism in intracellular redox signalling events (*Murphy, 2009*; *Murphy, 2012*). To monitor mitochondrial redox states at different times p.i. directly, we stained cells with a reversible sensor of mitochondrial ROS, flavin-rhodamine redox sensor 2 (FRR2) (*Kaur et al., 2016*) alongside Mitotracker Deep Red to visualise mitochondrial localization. The oxidised form of

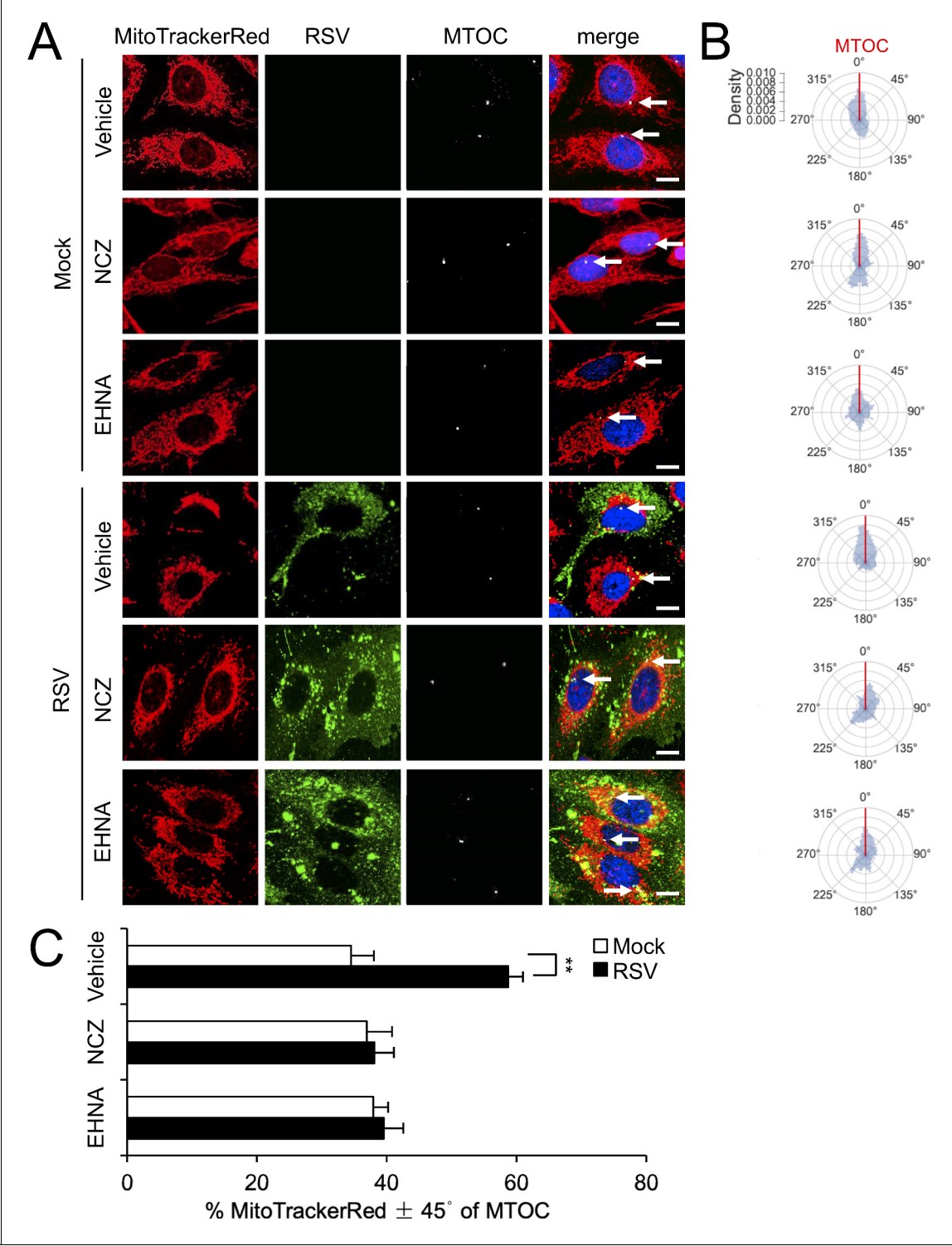

**Figure 4.** RSV-induced asymmetric distribution of mitochondria is microtubule- and dynein-dependent. (**A–C**) A549 cells were mock- or RSV-infected (MOI 1) for 18 hr with the indicated treatments over the last 2 hr as per **Figure 3AB**, followed by staining for mitochondria, RSV, MTOC and nuclei as per **Figure 2A**, and imaging by CLSM. (**A**) Merge panels overlay all four stains; arrows indicate the MTOC. In all panels, scale bar = 10 µm. (**B and C**) Polar kernel density of mitochondrial distribution relative to the MTOC analysed as per **Figure 2BC**. (**C**) Proportion of mitochondrial signal detected within 45° either side of the MTOC was quantitated. Results represent the mean ± SEM for n = 3 independent experiments, each of which analysed 25–30 cells per sample. **p<0.01.

DOI: https://doi.org/10.7554/eLife.42448.011

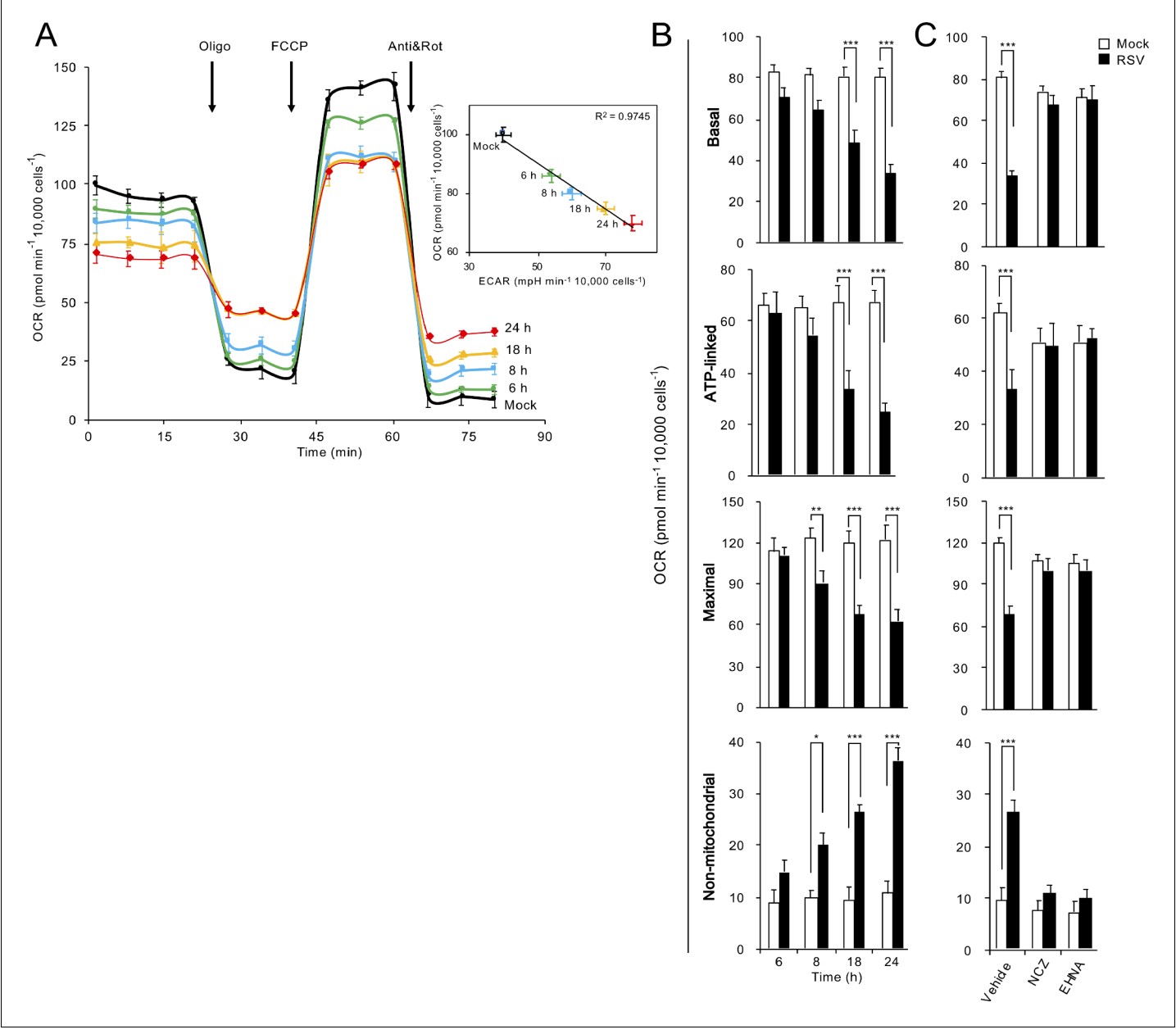

**Figure 5.** RSV infection inhibits host mitochondrial respiration in dynein/microtubule-dependent fashion. Cellular bioenergetic analysis was performed using the Seahorse XF96 Extracellular Flux Analyser. A549 cells were (**A and B**) mock-infected for 24 hr or RSV-infected (MOI 1) for 6–24 hr or (**C**) RSV-infected (MOI 1) for 18 hr with additions of the microtubule-depolymerizing agent nocodazole (NCZ, 17 µM), or the dynein ATPase-inhibitor EHNA (200 µM) over the last 2 hr. (**A**) An example of a typical oxygen consumption rate (OCR) obtained in these experiments. OCR was measured in real time upon sequential additions of ATP synthase inhibitor oligomycin (Oligo, 1 µM), proton ionophore FCCP (1 µM), mitochondrial complex III inhibitor antimycin A (Anti, 1 µM) and mitochondrial complex I inhibitor rotenone (Rot, 1 µM). *Inset:* Correlation of OCR, a measure of mitochondrial respiration and extracellular acidification rate (ECAR), an indicator of glycolysis ($R^2 = 0.9745$). (**B and C**) Mitochondrial respiration function parameters of basal, ATP-linked, maximal and non-mitochondrial respiration were determined as per *Figure 5—figure supplement 1B*. Results represent the mean ± SEM for n = 3 independent experiments, each performed in triplicate. ***$p<0.001$, **$p<0.01$, *$p<0.05$ compared to the mock-infected cells.
DOI: https://doi.org/10.7554/eLife.42448.012

The following figure supplement is available for figure 5:

**Figure supplement 1.** Correlation of multiplicity of infection (MOI) and effects on host cell mitochondrial respiration.
DOI: https://doi.org/10.7554/eLife.42448.013

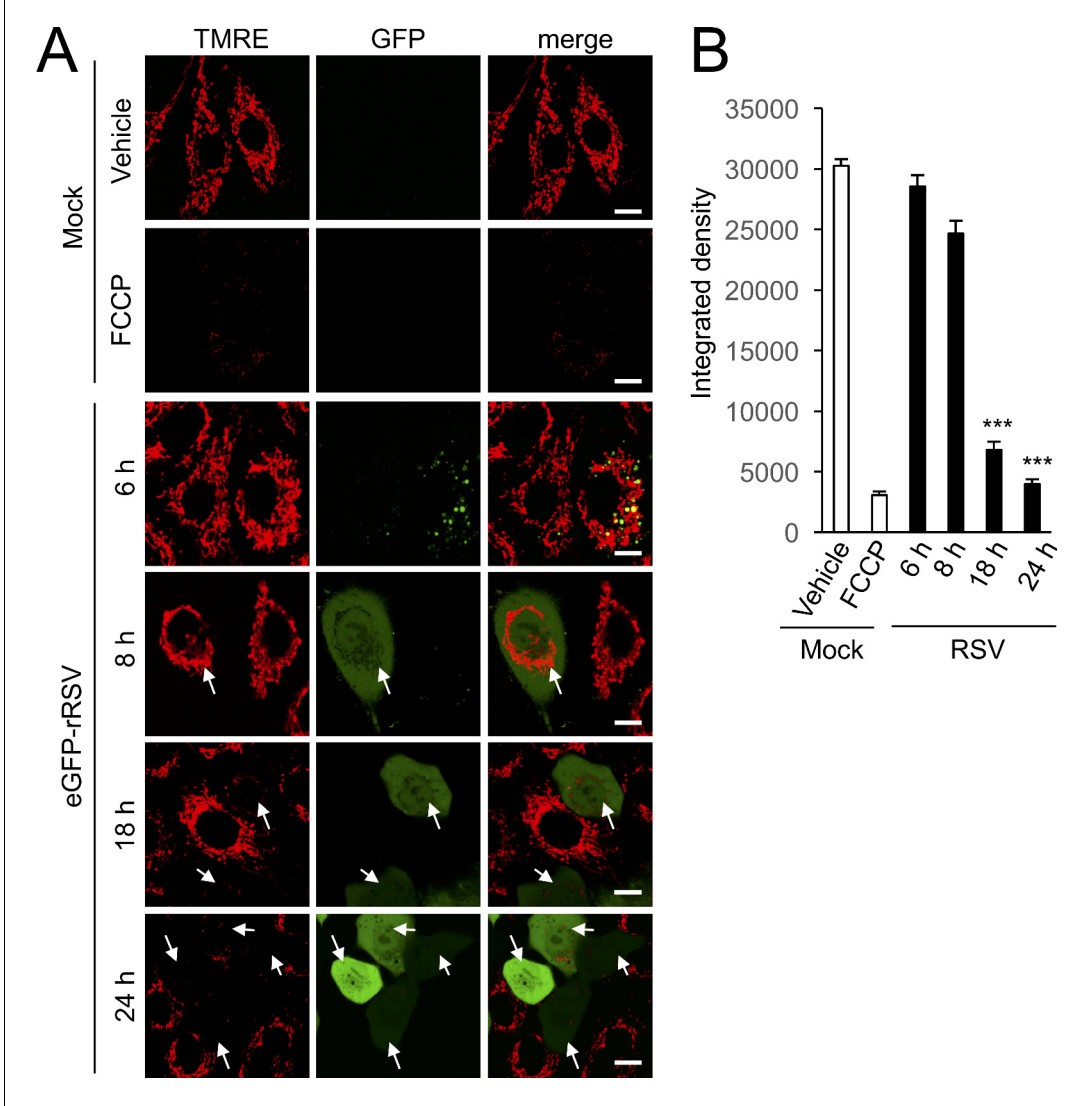

**Figure 6.** RSV infection disrupts maintenance of mitochondrial membrane potential ($\Delta\psi_m$). A549 cells were infected without (mock) or with eGFP-rRSV (MOI 1) for 6–24 hr, as indicated, followed by treatment with DMSO as vehicle or FCCP (5 μM, 10 min). In all cases, the $\Delta\psi_m$-sensitive dye tetramethylrhodamine ethyl ester (TMRE, red; 50 nM) was included for the final 15 min. (**A**) Cells were imaged live by CLSM. Merge panels overlay TMRE and GFP. Arrows indicate the eGFP-rRSV-infected cells. In all panels, scale bar is 5 μm. (**B**) Integrated intensity for TMRE fluorescence was quantified using Fiji software. Results represent the mean ± SEM for n = 3 independent experiments, each of which analysed 15–20 cells per sample ***p<0.001 compared to the mock-infected cells.

DOI: https://doi.org/10.7554/eLife.42448.014

The following figure supplement is available for figure 6:

**Figure supplement 1.** RSV infection impairs mitochondrial membrane potential ($\Delta\psi_m$).
DOI: https://doi.org/10.7554/eLife.42448.015

FRR2 emits at 580 nm much more strongly upon excitation at 514 nm than at 488 nm, further enabling ratiometric live imaging of mitochondrial ROS production in situ (*Kaur et al., 2016*). We treated mock-infected cells with rotenone as a positive control, with strong FRR2 emission in regions colocalizing with Mitotracker Deep Red (*Figure 7A*, 3rd row of panels), indicative of high levels of mitochondrial ROS. To confirm the mitochondrial contribution to this staining, we used the mitochondrial ROS scavenger mitoquinone mesylate (MitoQ) (*Smith and Murphy, 2010*; *Maharjan et al., 2014*), which strongly suppresses the actions of rotenone (*Figure 7A*, 4th row of panels). FRR2 fluorescence increased in RSV-infected cells, but was also reduced by MitoQ

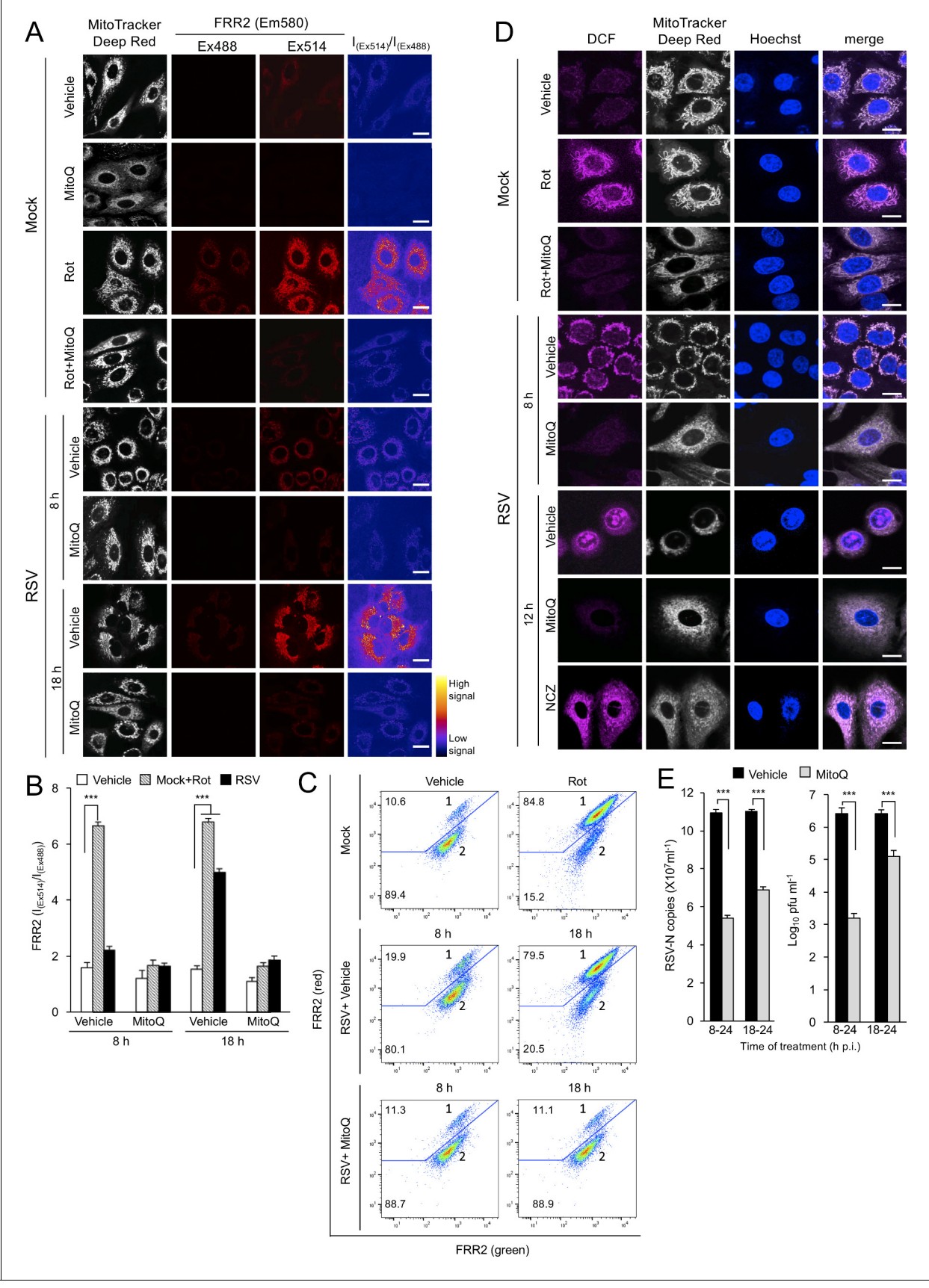

**Figure 7.** RSV infection enhances mitochondrial reactive oxygen species (ROS) generation, which favours virus production. (A–D) A549 cells were mock-infected or RSV-infected (MOI 3) for 8, 12 or 18 h p.i., with the the additions as indicated for the final 2 hr prior to staining: the mitochondrial complex I inhibitor rotenone (Rot, 0.5 µM), the mitochondria-specific ROS scavenger mitoquinone mesylate (MitoQ, 1 µM), NCZ (17 µM) or DMSO as a vehicle; in the case of dual Rot/MitoQ addition (Rot + MitoQ), Rot was added 4 hr before staining. (A) Cells were stained for Mitotracker Deep Red (white; 100 nM, 15 min) and the mitochondria-specific ROS probe flavin-rhodamine redox sensor 2 (FRR2, red; 2 µM, 15 min prior to imaging). Colocalization for FRR2 staining at either Ex488 or Ex514 and Mitotracker Deep Red was >85% (Pearson correlation coefficient; *Costes et al., 2004*) across all samples (25–30 cells/sample). The ratiometric output images of $I_{(Ex514)}/I_{(Ex488)}$ (far right) were calculated by pixelwise division of FRR2 emission (580 ± 20 nm) images acquired using excitation at 514 nm (third column) or 488 nm (second column), and are represented in pseudo-colour (intensity colour key displayed lower right). Live cell imaging was performed by resonant scanning CLSM. Results are typical of 3 independent experiments. In all panels, scale bar = 10 µm. (B) FRR2 ($I_{(Ex514)}/I_{(Ex488)}$) was calculated for the mitochondrial regions defined by Mitotracker Deep Red staining in the $I_{(Ex514)}/I_{(Ex488)}$ images such as those in (A) using a custom CellProfiler pipeline (see Materials and methods). Results represent the mean ± SEM for n = 3 independent experiments, where each experiment analysed 25–30 cells per sample, ***p<0.001. (C) FACS analysis from single-cell suspensions stained with FRR2. Green (540/30 nm) and red (585/42 nm) fluorescence was excited at 488 nm. The percentages of cells (50,000/sample) in populations 1 (high red emission) and 2 (low red emission) determined using FlowJo are indicated. Results were typical of 3 independent experiments. (D) Cells were stained with Mitotracker Deep Red as for (A), Hoechst nucleic acid dye (blue; 5 µg/ml) and the cellular ROS indicator 2′,7′-dichlorodihydrofluorescein diacetate (DCF, magenta; 2.5 µM) over the last 5 min before live cell imaging by CLSM. Merge panels overlay all three stains. In all panels, scale bar = 10 µm. (E) Mock or RSV-infected (MOI 1) A549 cells were treated with MitoQ (1 µM) for the times indicated, followed by cell lysate preparation, with qPCR and plaque assay performed to determine viral RNA copy number and infectious virus (plaque forming units pfu ml$^{-1}$) respectively. Results shown represent the mean ± SEM from three independent experiments assayed in triplicate. ***p<0.001.

DOI: https://doi.org/10.7554/eLife.42448.016

The following figure supplement is available for figure 7:

**Figure supplement 1.** The mitochondria-targeted antioxidant MitoQ suppresses RSV replication but not virus spread.

DOI: https://doi.org/10.7554/eLife.42448.017

(*Figure 7A*; bottom 4 rows of panels). Ratiometric visualization ($I_{(Ex514)}/I_{(Ex488)}$, right column) indicated that the punctate source of ROS causing FRR2 oxidation corresponded to mitochondria revealed by Mitotracker Deep Red staining (*Figure 7A*). Quantitative analysis of the ratiometric images highlighted the extent of ROS production (*Figure 7B*), with the results indicating that RSV infection increases mitochondrial ROS generation significantly (p<0.001) by 18 h p.i.

Flow cytometric analysis can exploit the green and red fluorescence emission by the oxidised form of FRR2 (*Kaur et al., 2016*). When we analysed cell suspensions from FRR2-stained mock- and RSV-infected cells, two distinct populations could be discerned: population one showing both higher red and green emission indicative of FRR2 oxidation through higher mitochondrial ROS generation and population two with lower emission (*Figure 7C*). Compared to mock infection, RSV infection at 8 and 18 hr increased population 1 by about 2- and 8-fold, respectively, with these increases reversed by MitoQ (*Figure 7C*). Taken together, the results reveal that RSV infection elevates mitochondrial ROS generation.

To visualise intracellular ROS production directly in RSV infection, we stained RSV-infected cells at different times p.i with the intracellular ROS indicator 2′,7′-dichlorodihydrofluorescein diacetate (DCF) (*Al-Mehdi et al., 2012*), again alongside Mitotracker Deep Red to enable visualization of mitochondrial localization, and also the Hoechst dye to define cell nuclei. We treated mock-infected cells with rotenone as a positive control, with strong DCF staining in regions colocalized with Mitotracker Deep Red (*Figure 7D*, 2$^{nd}$ row) indicating high levels of mitochondrial ROS. Analysis of RSV-infected cells revealed higher levels of ROS associated with perinuclear mitochondria at 8 h p.i.; this staining at 12 h p.i. was still perinuclear, but with intense nuclear staining as well as some diffuse cytoplasmic staining (*Figure 7D*). To assess the mitochondrial contribution to this ROS staining, we confirmed the actions of MitoQ to suppress the actions of rotenone (*Figure 7D*, 3$^{rd}$ row). We then treated RSV-infected cells with MitoQ 2 hr before imaging and observed a marked reduction in DCF staining, suggesting that the ROS generated in the absence of MitoQ in infected cells was largely mitochondrial (*Figure 7D*). Strikingly, when we included nocodazole treatment to prevent microtubule-dependent mitochondrial clustering upon RSV infection, DCF staining remained largely mitochondrial and cytoplasmic and was largely excluded from the nucleus (*Figure 7D*, bottom row). An implication of these findings is that mitochondrial perinuclear redistribution is a prerequisite for altering the oxidative status of the host cell nucleus during RSV infection.

To confirm the physiological relevance of the above results with respect to RSV-stimulated mitochondrial ROS production in the context of the RSV infectious cycle, MitoQ was tested for its ability

to inhibit virus production in A549 cells (*Figure 7E*). Excitingly, addition of MitoQ at 8 or even 18 h p.i. had a significant (p<0.001) inhibitory effect, reducing viral genome replication by up to 60% and infectious virus production by up to 3.5 logs measured at 24 h p.i. (*Figure 7E*). These results confirm that mitochondrial ROS production contributes essentially to RSV virus production, and that MitoQ, as a specific inhibitor of mitochondrial ROS, is a potent inhibitor of RSV infection.

To test if mitochondrial ROS generation contributes to the spread of RSV infection through cell fusion, we tracked syncytia (multinucleated, fused cells) formation in RSV-infected (MOI 0.3) Vero cells over 24–48 h p.i. in the presence or absence of MitoQ (*Figure 7—figure supplement 1A*). MitoQ was found not to alter the density or size of syncytia formed at 48 h p.i. (*Figure 7—figure supplement 1B*), suggesting that mitochondrial ROS generation facilitates infectious virus production by promoting viral replication rather than enhancing virus spread to neighboring uninfected cells through cell fusion.

## Mitochondrial redistribution is key to RSV-induced mitochondrial ROS generation.

Previous studies have indicated that microtubule and dynein are necessary for the formation of RSV infectious virus filaments (*Vanover et al., 2017*). To confirm the physiological significance of RSV-induced mitochondrial redistribution, we tested the effect on RSV infectious virus production in A549 cells of agents inhibiting mitochondrial distribution and/or ROS production. Treatment with the microtubule/dynein motor targeting agents nocodazole or EHNA significantly (p<0.05) reduced (up to 2.5 log) infectious virus production (*Figure 8—figure supplement 1A*), in contrast to the actin-targeted agent cytochalasin D or the kinesin-inhibitor monastrol that had no significant effect. Similarly, siRNA-knockdown of proteins of the dynein complex (*DYNLT1* or *DYNC1H1*), but not the kinesin complex (*KLC1*), resulted in significant (p<0.001) decreases in infectious virus production (2–3 logs) compared to the scrambled siRNA control (*Figure 8—figure supplement 1B*).

None of the treatments above impacting mitochondrial distribution induced by RSV infection, apart from nocodazole, impact microtubules directly (see *Figure 3AD*), implying that mitochondrial redistribution, rather than the microtubule network per se, is key to RSV infection. To examine this idea further, we performed knockdown experiments for the mRNA-binding protein *CLUH* (clustered mitochondria homolog) (*Figure 8A*), which plays a key role in mitochondrial distribution independent of the microtubule network by facilitating translation of nuclear-encoded mitochondrial genes close to mitochondria (*Wakim et al., 2017*; *Gao et al., 2014*). As observed previously (*Wakim et al., 2017*; *Gao et al., 2014*), *CLUH*-targeting siRNA but not control scrambled siRNA induced mitochondrial perinuclear clustering in mock infected cells (*Figure 8B*; first two rows), with RSV-infection resulting in only a further slight increase in clustering (*Figure 8B*; 4th row). This was confirmed by quantitative analysis for the $R_{90\%}$ parameter (*Figure 8C*). Strikingly, the *CLUH* knockdown-induced mitochondrial redistribution resulted in elevated mitochondrial ROS production, as indicated by strong FRR2 emission in regions colocalizing with Mitotracker Deep Red as revealed by ratiometric live cell imaging in mock-infected cells (*Figure 8D*; first two rows); RSV infection further enhanced the effect (*Figure 8D*; last two rows). Quantitative analysis of the ratiometric images confirmed the results, indicating that depletion of *CLUH* significantly (p<0.01) increased mitochondrial ROS levels in mock- and RSV-infected cells (*Figure 8E*).

Finally, the physiological relevance of these results with respect to the RSV infectious cycle could be confirmed by showing that siRNA-mediated *CLUH* knockdown resulted in significantly (p<0.01) increased virus production (*Figure 8—figure supplement 1C*), The results overall confirm that mitochondrial reorganization, rather than the microtubule network per se, is key to RSV infection, through its link to mitochondrial ROS production.

To confirm that the various drug and siRNA treatments used above limiting RSV infection do not do so simply by impacting cell viability, we assessed release of the cytosolic enzyme lactate dehydrogenase (LDH) into the culture medium, indicative of cell death. None of our drug (*Figure 8—figure supplement 2A*) or siRNA (*Figure 8—figure supplement 2B*) treatments significantly increased LDH release compared to that of controls in either the absence or presence of infection, consistent with the idea that all of the treatments affecting RSV virus production did so through specific effects on mitochondrial distribution/mitochondrial ROS production, rather than as a result of toxicity.

Together, the results indicate that mitochondrial ROS generation facilitates RSV infection, with RSV infection effectively co-opting mitochondria in a microtubule- and dynein-dependent fashion to

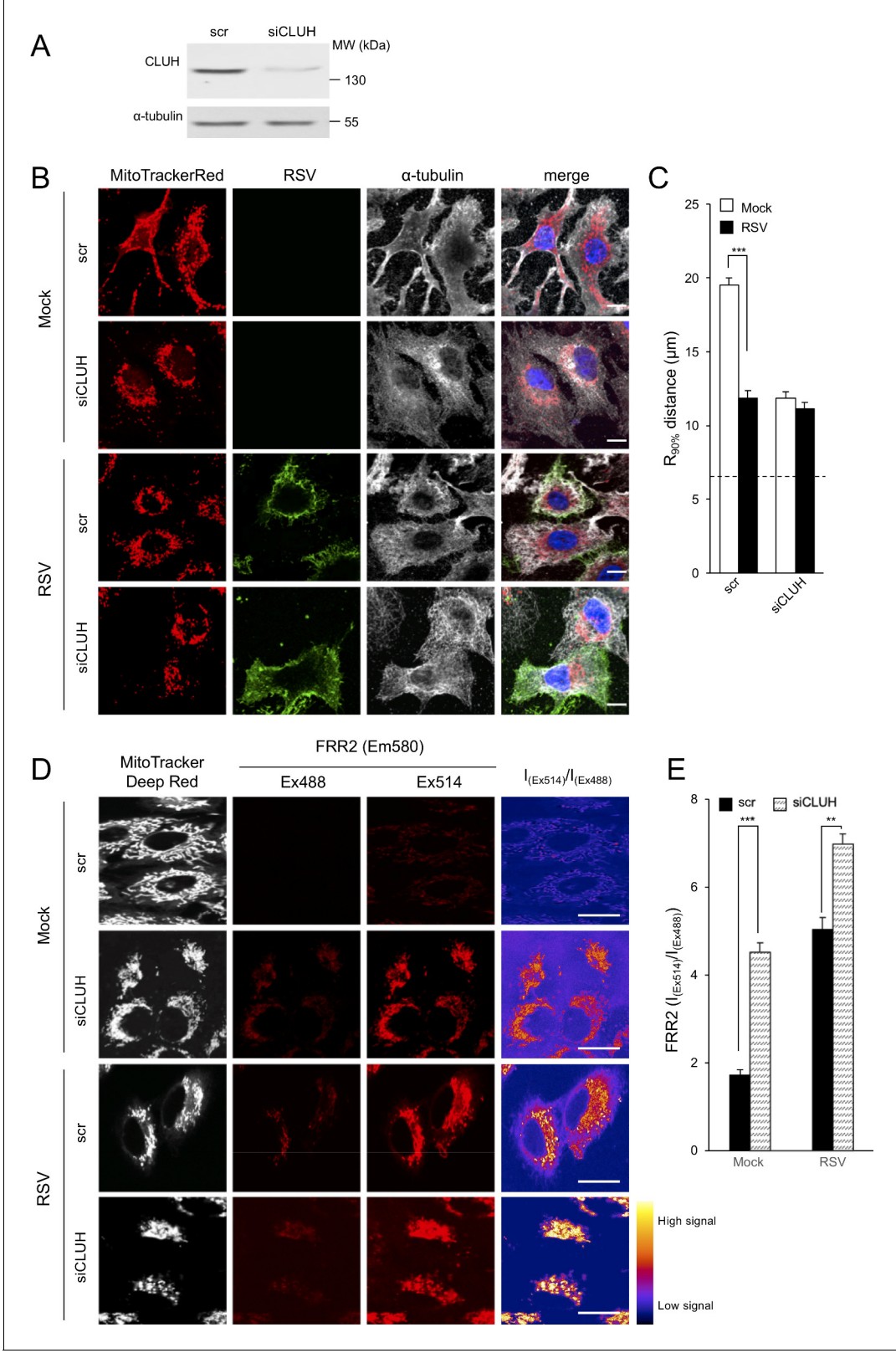

**Figure 8.** Knockdown of the clustered mitochondria homolog (CLUH) elicits perinuclear mitochondrial redistribution, elevated mitochondrial ROS generation, and enhances RSV virus production. (A–E) A549 cells were pretreated (48 hr) with siRNA (50 nM) for clustered mitochondria homolog (*CLUH*), or scrambled siRNA control (scr) followed by infection with RSV (MOI 1) for another 24 hr. (A) Immunoblot analysis for *CLUH*, or the control α-tubulin, as indicated (40 μg cell lysate protein/lane). (B and C) Immunostaining and R$_{90\%}$ analysis were as per (*Figure 1C and D*). In all panels, scale

*Figure 8 continued on next page*

*Figure 8 continued*

bar = 10 µm. Results represent the mean ± SEM for n = 3 independent experiments, where each experiment analysed 25–30 cells per sample; ***p<0.001. (D) Cells were stained with Mitotracker Deep Red and the mitochondria-specific ROS probe FRR2 as per *Figure 7A*. The ratiometric output images of $I_{(Ex514)}/I_{(Ex488)}$ were calculated as per *Figure 7A*. (E) FRR2 ($I_{(Ex514)}/I_{(Ex488)}$) was calculated for the mitochondrial regions defined by Mitotracker Deep Red staining in the $I_{(Ex514)}/I_{(Ex488)}$ images such as those in D) using a custom CellProfiler pipeline as per *Figure 7B*. Results represent the mean ± SEM for n = 3 independent experiments, where each experiment analysed 25–30 cells per sample, ***p<0.001, **p<0.01.

DOI: https://doi.org/10.7554/eLife.42448.018

The following figure supplements are available for figure 8:

**Figure supplement 1.** Microtubule/dynein-dependent mitochondrial redistribution favours RSV virus production.

DOI: https://doi.org/10.7554/eLife.42448.019

**Figure supplement 2.** Lack of effect of drug (A) and siRNA (B) treatments used in this study on cell viability.

DOI: https://doi.org/10.7554/eLife.42448.020

**Figure supplement 3.** Mitochondrial ROS-dependent Effects on RSV Viral replication and infectious virus production in primary human bronchial epithelial cells (pBECs).

DOI: https://doi.org/10.7554/eLife.42448.021

---

perinuclear/asymmetric mitochondrial distribution that favours reduced mitochondrial respiration and enhanced mitochondrial ROS production. Blocking RSV induced reorganization of host cell mitochondria and increased mitochondrial ROS production thus inhibits RSV infection effectively.

## MitoQ protects against RSV infection in primary human bronchial epithelial cell and mouse models

The results for the A549 human alveolar line indicating that the mitochondrial ROS scavenger MitoQ can be a potent inhibitor of RSV infection (*Figure 7E*) were firstly extended by using primary human bronchial epithelial cells (pBECs) infected with RSV followed by the addition of DMSO (vehicle) or the mitochondrial ROS scavenger MitoQ for the last 18 hr, prior to assessment of viral replication and infectious virus production (*Figure 8—figure supplement 3*). Whereas RSV infection of the vehicle-treated cells resulted in high levels of viral genomes (*Figure 8—figure supplement 3A*) and infectious virus titres (*Figure 8—figure supplement 3B*) at 18 and 36 h p.i., cells treated with MitoQ showed significantly (p<0.001) reduced numbers of viral genomes (*Figure 8—figure supplement 3A*;>70%) and infectious virus titres (*Figure 8—figure supplement 3B*;>4 logs) at both timepoints p.i. These results were consistent with the idea that elevated mitochondrial ROS is critical for RSV infection in a clinically relevant human infectious model.

To confirm the contribution of mitochondrial ROS generation to disease in the affected lung in vivo, we used the established BALB/c mouse model of RSV infection (*van Schaik et al., 1998*; *Taylor et al., 1984*). As previously (*van Schaik et al., 1998*; *Taylor et al., 1984*), viral replication and infectious virus titres peaked at day 5 p.i., declining rapidly at day 7 p.i. (*Figure 9AB*, black bars). Mice treated with MitoQ (*Figure 9AB*, grey bars) showed significantly (p<0.001) reduced (>4 fold) viral replication and infectious virus production compared to the vehicle control for days 4–7 p.i. Importantly, MitoQ treatment resulted in significantly (p<0.001) less dense inflammatory cell infiltrate (characterised by mononuclear cells and eosinophils) around the bronchial airways (*Figure 9CD*) and perivascular regions (*Figure 9EF*) of the lungs, as revealed by histological sectioning and objective blind assessment (*Ford et al., 2001*; *Mehra et al., 2012*). Consistent with the alleviated host response to infection in the case of MitoQ treatment, we observed significantly (p<0.01) reduced levels of systemic RANTES, a chemokine highly chemoattractant for the inflammatory infiltrate, upon MitoQ administration throughout (*Figure 9G*). Taken together, MitoQ treatment suppresses RSV infection and decreases virus-induced inflammation in mice, with clear therapeutic implications.

## Discussion

This study shows for the first time that RSV infection co-opts host cell mitochondria to favour infection; over an 8–24 hr period of infection, RSV progressively impacts the host cell, with mitochondrial redistribution to a perinuclear location near the MTOC, decreased mitochondrial respiration, a loss of $\Delta\psi_m$, and increased mitochondrial ROS generation (*Figure 10*). These events are dependent on

host cell microtubule integrity and dynein, implying that this RSV-induced mitochondrial redistribution is enacted via a dynein-driven/retrograde-directed mode of transport that is central to the effects on the host cell; our observations that altering microtubule integrity or dynein activity can block effects of infection on mitochondrial function as well as RSV infectious virus production reiterate that these events are critical to RSV infection (*Figure 10*). Interestingly, knockdown of the mitochondrial biogenesis factor CLUH appears to be able to further enhance mitochondrial ROS production in RSV infection (*Figure 8*) and effect a boost in infectious virus production (*Figure 8—figure supplement 1C*), implying that CLUH, in contrast to dynein, functions to limit virus-induced perinuclear clustering (*Figure 10*). The most striking observation, however, is that the mitochondrial ROS scavenger MitoQ can markedly reduce viral replication and infectious virus production (*Figure 7E*) as well as restore mitochondrial distribution during infection (*Figure 7D*), clearly implicating RSV-enhanced mitochondrial ROS production as a key contributor to the infectious process. Most importantly, blocking mitochondrial ROS generation significantly reduced viral replication/production and the extent of lung and systemic inflammation in a mouse model (*Figure 9*), highlighting the clinical relevance of our findings.

Oxidative stress is known to play a fundamental role in the pathogenesis of RSV-associated lung inflammatory disease, correlating strongly with disease severity (*Hosakote et al., 2009*; *Castro et al., 2006*). The mechanism by which elevated ROS contributes to RSV infection may in part relate to effects at the level of the nucleus, with nuclear ROS (eg. see *Figure 7D*) impacting host nuclear gene transcription (*Munday et al., 2015*; *van Diepen et al., 2010*; *Kipper et al., 2015*). Significantly, progressive increases in lipid peroxidation products in parallel with lowered reduced glutathione levels in RSV-infected airway epithelial cells indicate the increased oxidative stress in cells following RSV infection (*Hosakote et al., 2009*). Antioxidants have been reported to limit RSV infection in cell culture as well as in mouse models (*Hosakote et al., 2009*; *Castro et al., 2006*; *Komaravelli et al., 2015*; *Zang et al., 2011*). This study shows for the first time that treatment with an agent specifically scavenging mitochondrial ROS can limit viremia, significantly reduce levels of RANTES chemokine and ameliorate lung and systemic inflammation (*Figure 9*), implying that modulation of oxidative stress in the context of RSV infection can help diminish lung disease. This is consistent with the work of *Castro et al. (2006)*, who showed that the antioxidant butylated hydroxyanisole (BHA) can reduce levels of chemokine (RANTES) and lung inflammation in RSV-infected mice, as well as the fact that antioxidants have been reported to help alleviate symptoms in paediatric patients with clinical RSV infection (*Dowell et al., 1996*; *Kawasaki et al., 1999*). Thus, mounting evidence highlights the potential of antioxidants to attenuate symptoms and pathology in RSV infection.

Importantly, the present study uncovers several novel impacts of RSV infection on host cell mitochondria that are relevant to future therapeutic approaches. Specifically, the study shows for the first time that agents that inhibit microtubule-/dynein-dependent mitochondrial redistribution and/or reduce mitochondrial ROS limit RSV infection, with MitoQ in particular able to decrease viremia and airway inflammation in mice. Significantly, MitoQ has been safely delivered in oral form to patients for up to a year as indicated by two phase II clinical trials (*Smith and Murphy, 2010*; *Maharjan et al., 2014*); clearly, the findings introduce the possibility of using MitoQ or a similar antioxidant as an effective anti-RSV agent. Therapeutic modulation of host cell mitochondrial ROS production thus presents itself as an exciting possibility to counteract RSV infection.

## Materials and methods

**Key resources table**

| Reagent type (species) or resource | Designation | Source or reference | Identifiers | Additional information |
|---|---|---|---|---|
| Strain (*Respiratory Syncytial Virus*) | RSV A2 strain | PMID: 27464690 | | |

*Continued on next page*

*Continued*

| Reagent type (species) or resource | Designation | Source or reference | Identifiers | Additional information |
|---|---|---|---|---|
| Strain, Strain background (*Respiratory Syncytial Virus*) | eGFP-rRSV | PMID: 24418538 | Gift from Michael N Teng, University of South Florida | |
| Mouse strain, (*M. musculus*) | BALB/c | The Jackson Laboratory | #: 000651 | |
| Genetic reagent (*H. sapiens*) | DYNLT1 (siRNA) | GE Dharmacon, SMART pool | #: 6993 | Used for transfection (50 nM) |
| Genetic reagent (*H. sapiens*) | DYNC1H1 (siRNA) | GE Dharmacon, SMART pool | #: 1778 | Used for transfection (50 nM) |
| Genetic reagent (*H. sapiens*) | KLC1 (siRNA) | GE Dharmacon, SMART pool | #: 3831 | Used for transfection (50 nM) |
| Genetic reagent (*H. sapiens*) | CLUH (siRNA) | GE Dharmacon, SMART pool | #: 23277 | Used for transfection (50 nM) |
| Cell line (*H. sapiens*) | A549 | ATCC | CCL-185 | Mycoplasma tested and/or STR profiled |
| Cell line (*Chlorocebus sp.*) | Vero | ATCC | CCL-81 | |
| Cell line (*H. sapiens*) | BCi-NS1 | PMID: 24298994 | Provided by Alan Hsu, Philip M Hansbro, and Peter AB Wark, University of Newcastle | |
| Primary Cells (*H. sapiens*) | pBECs | PMID: 15781584 | Provided by Alan Hsu, Philip M Hansbro, and Peter AB Wark, University of Newcastle | |
| Antibody | Goat polyclonal anti-RSV | Abcam | Cat. #: ab20745, RRID:AB_777677 | IF (1:400) |
| Antibody | Mouse monoclonal anti-γ-tubulin | Proteintech | Cat. #: 66320–1-Ig | IF (1:300) |
| Antibody | Mouse monoclonal anti-α-tubulin | Santa Cruz | Cat. #: sc-5286, RRID: AB_628411 | IF (1:100), WB (1:5000) |
| Antibody | Rabbit polyclonal anti-Goat IgG, Secondary Antibody, Alexa Fluor 488 | ThermoFisher Scientific | Cat. #: 11078 RRID: AB_2534122 | IF (1:1000) |
| Antibody | Donkey polyclonal anti-Mouse IgG, Secondary Antibody, Alexa Fluor 647 | ThermoFisher Scientific | Cat. #: A-31571, RRID: AB_162542 | IF (1:1000) |
| Antibody | Alexa Fluor 488 phalloidin | ThermoFisher Scientific | Cat. #: A12379, RRID: AB_2315147 | IF (1:1000) |
| Antibody | Mouse monoclonal anti-DYNC1H (C-5) | Santa Cruz | Cat. #: sc-514579 | WB (1:500) |

*Continued on next page*

*Continued*

| Reagent type (species) or resource | Designation | Source or reference | Identifiers | Additional information |
|---|---|---|---|---|
| Antibody | Mouse monoclonal anti-DYNLT1 (H-11) | Santa Cruz | Cat. #: sc-365567, RRID: AB_10841719 | WB (1:1000) |
| Antibody | Mouse monoclonal anti-KLC1 (L2) | Santa Cruz | Cat. #: sc-58776, RRID: AB_784214 | WB (1:1000) |
| Antibody | Rabbit polyclonal anti-CLUH | ThermoFisher Scientific | Cat. #: PA5-71324, RRID: AB_2690757 | WB (1:1000) |
| Antibody | Goat polyclonal anti-Mouse IgG, HRP Conjugate Antibody | Promega | Cat. #: W4021, RRID: AB_430834 | WB (1:10,000) |
| Recombinant DNA reagent | CellLight Golgi-GFP *BacMam 2.0* | ThermoFisher Scientific | Cat. #: C10592 | |
| Recombinant DNA reagent | CellLight Mitochondria-RFP *BacMam 2.0* | ThermoFisher Scientific | Cat. #: C10601 | |
| Commercial assay or kit | Mouse RANTES ELISA Kit | RayBiotech Inc | Cat. #: ELM-RANTES | |
| Commercial assay or kit | LDH Cytotoxicity Detection Kit | Roche Applied Science | Cat. #: 11644793001 | |
| Commercial assay or kit | TMRE $\Delta\psi_m$ Assay Kit | Abcam | Cat. #: ab113852 | Live-cell IF (50 nM) |
| Chemical compound, drug | Oligomycin | Seahorse XF Cell Mito Stress Test Kit | Cat. #: 103015–100 | SBA (1 µM) |
| Chemical compound, drug | FCCP | Seahorse XF Cell Mito Stress Test Kit | Cat. #: 103015–100 | SBA (1 µM) |
| Chemical compound, drug | Antimycin A | Seahorse XF Cell Mito Stress Test Kit | Cat. #: 103015–100 | SBA (1 µM) |
| Chemical compound, drug | Rotenone | Seahorse XF Cell Mito Stress Test Kit | Cat. #: 103015–100 | SBA (1 µM) |
| Chemical compound, drug | MitoQ | Health Manufacturing, New Zealand | Gift from Health Manufacturing, New Zealand | PA (0.5 µM), IF (1 µM) |
| Chemical compound, drug | Nocodazole | Sigma | Cat. #: M1404 | PA and IF (17 µM) |
| Chemical compound, drug | EHNA | Sigma | Cat. #: E114 | PA and IF (200 µM) |
| Chemical compound, drug | Monastrol | Sigma | Cat. #: M8515 | PA and IF (50 µM) |
| Chemical compound, drug | Cytochalasin D | Sigma | Cat. #: C8273 | PA and IF (2 µM) |
| Software, algorithm | Application Suite Advanced Fluorescence Lite | Leica | RRID:SCR_013673 | Version: 2.8.0, Build: 7266 |

*Continued*

| Reagent type (species) or resource | Designation | Source or reference | Identifiers | Additional information |
|---|---|---|---|---|
| Software, algorithm | ZEN 2 | Zeiss | RRID:SCR_013672 | Blue edition |
| Software, algorithm | Fiji | Fiji (https://fiji.sc/) | RRID:SCR_002285 | Version 2.0.0-rc-64, Build: e0512e3c19 |
| Software, algorithm | Custom Scripts for Quantitative Analysis of Mitochondrial Distribution | Programmed in Python | This paper | Quantitative analyses of mitochondrial organization can be accessed via https://gitlab.erc.monash.edu.au/mmi/mito (*Schulze, 2018*; copy archived at https://github.com/elifesciences-publications/mito) |
| Software, algorithm | FlowJo | Tree Star, Inc (http://www.flowjo.com) | RRID:SCR_000410 | Version 10.5.3 |
| Software, algorithm | GraphPad Prism | GraphPad Prism (https://graphpad.com) | RRID:SCR_015807 | Version 6 |
| Software, algorithm | CellProfiler | Broad Institute Inc (http://cellprofiler.org/) | RRID:SCR_007358 | Version 3.1.8 |
| Other | DAPI stain | Sigma | Cat. #: 10236276001 | IF (1:15000) |
| Other | MitoTrackerRed CMXRos | ThermoFisher Scientific | Cat. #: M7512 | IF (100 nM) |
| Other | Hoechst stain (33342) | ThermoFisher Scientific | Cat. #: H3570 | Live-cell IF (1:2000) |
| Other | TPE-Ph-In | PMID: 26264419 | Gift from Yuning Hong, La Trobe University | Live-cell IF (1:2 μM) |
| Other | DCF | ThermoFisher Scientific | Cat. #: D399 | Live-cell IF (2.5 μM) |
| Other | Mitotracker Deep Red | ThermoFisher Scientific | Cat. #: M22426 | Live-cell IF (100 nM) |
| Other | FRR2 | PMID: 26865422 | Gift from Jacek L Kolanowski and Elizabeth J New, the University of Sydney | Live-cell IF (2 μM) |
| Other | Biomeda Gel Mount | ProSciTech | Cat. #: EMS17985-11 | |
| Other | DharmaFECT siRNA transfection reagent | GE Dharmacon | Cat. #: T-2001–04 | |

IF - immunofluorescence, PA- plaque assay, SBA - Seahorse Bioenergetics Analysis, WB - Western blot.

## Cell culture, RSV infection and RSV growth

All cells/cell lines were confirmed mycoplasma free by regular testing. They were maintained in a humidified atmosphere (5% $CO_2$, 37°C) and passaged (3 day intervals) by dissociation with trypsin-EDTA (Gibco). A549 cells (human adenocarcinoma alveolar basal epithelial cells) were grown in Ham's F-12K (HF-12K) medium containing 2 mM L-glutamine (Gibco), 1.5 gl$^{-1}$ sodium bicarbonate, 10% heat-inactivated fetal calf serum (FCS; DKSH Australia Pty Ltd) and 100 U ml$^{-1}$ penicillin and streptomycin (Gibco). Vero cells (African green monkey kidney epithelial cells) were grown in Dulbecco's modified Eagle's medium (DMEM, Gibco) containing 2% heat-inactivated FCS. A549 (ATCC: CCL-185) and Vero (ATCC: CCL-81) lines have been verified by STR profiling.

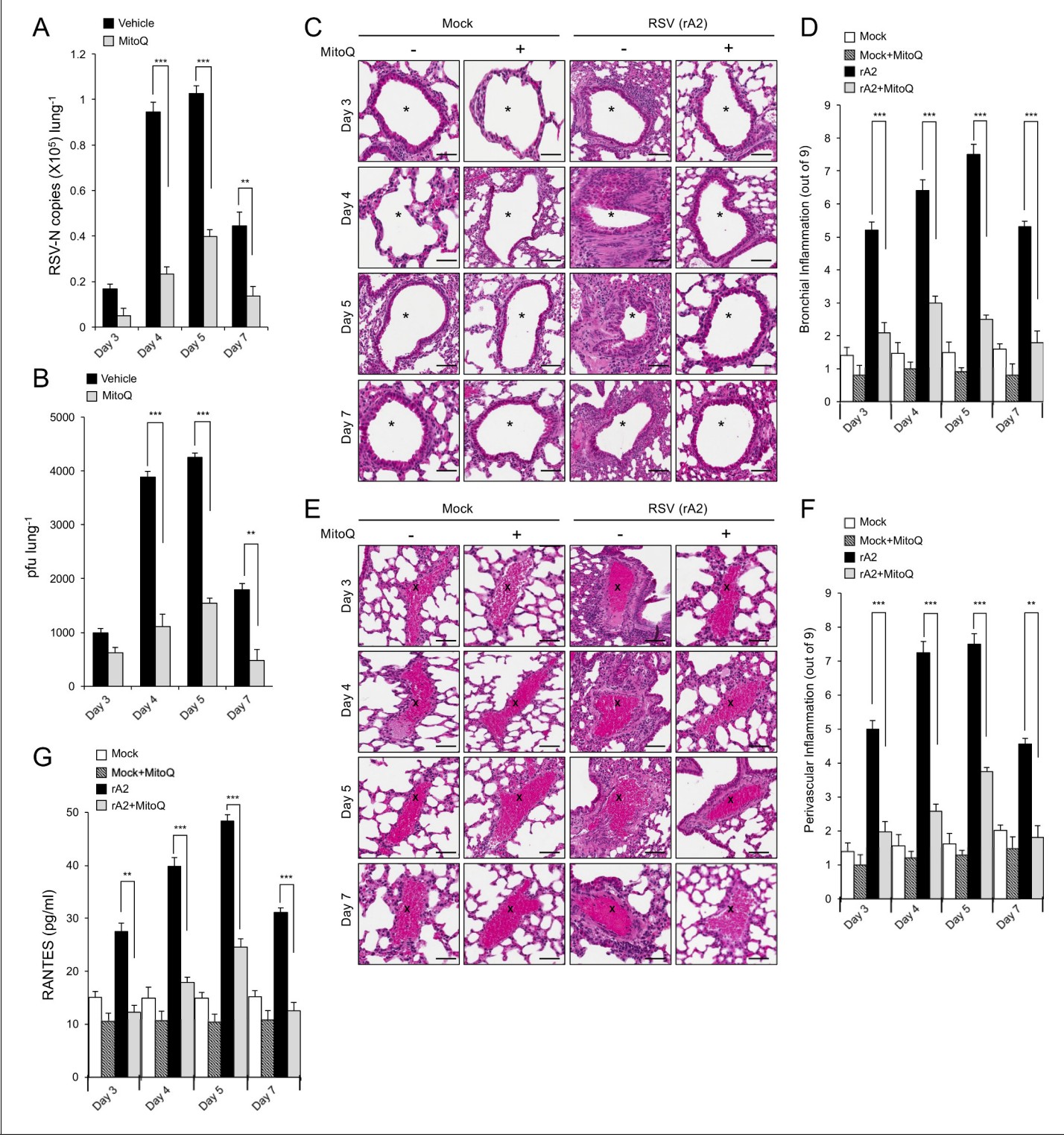

**Figure 9.** The mitochondrial ROS scavenger MitoQ reduces RSV virus production in mice, concomitant with reduced inflammation. BALB/c mice were intranasally inoculated with 50 μl of $2.5 \times 10^5$ pfu of rRSV or an equivalent volume of diluent at day 0 and given water containing MitoQ (500 μM) or fresh water *ad libitum* each day. five mice from each treatment group (total of 20 mice) were euthanased on the days p.i. indicated, and samples collected for analysis. (**A and B**) One lung from each animal was put into 1 mL of diluent with steel beads and frozen at −80˚C. Lungs were subsequently homogenised in a tissue-lyser, debris removed by centrifugation and the supernatant used immediately for (**A**) virus genome analysis by qPCR and (**B**) quantification of infectious virus (plaque forming units or pfu/lung) by plaque assay. (**C–F**) The other lung was fixed in formalin, embedded in paraffin, sectioned and stained with haemotoxylin and eosin (H and E). (**C and E**) Representative images (enlarged 400X) are from whole

*Figure 9 continued on next page*

*Figure 9 continued*

lung sections scanned using Aperio ScanScope slide scanner. Scale bar = 100 μm. Inflammatory cell infiltrate surrounds the bronchial airway (marked by asterisks, (C) or the blood vessels (marked by crosses, (E) in rRSV-infected samples. Each lung was scanned at three different depths. (D and F) Pooled data for scoring the extent of (D) bronchial and (F) perivascular inflammation. The intensity of inflammation was quantitated double-blind according to the schema described in Materials and methods (0–9 scale). Quantitation was performed on multiple lung lobes from three different depths of sectioned tissue. Results represent the mean ± SEM (n = 15). (G) Blood was collected by cardiac puncture. Systemic inflammation was determined by ELISA for RANTES as described in Materials and methods. Results represent the mean ± SEM (n = 5). ***p<0.001, **p<0.01.

DOI: https://doi.org/10.7554/eLife.42448.022

Immortalised-non-smoker one basal cells (BCi-NS1) verified by karyotyping were sourced directly from *Walters et al. (2013)* and grown in Bronchial Epithelial Growth Media (BEGM, Lonza). Primary human bronchial epithelial cells (pBECs) were obtained from four healthy individuals who had no history of smoking or lung disease, had normal lung function, and gave written, informed consent to participate and have their data published, in accordance with the procedures approved by the University of Newcastle Human Ethics Committee (Project Ref. No. H-163–1205), in keeping with the guidelines of the National Institutes of Health, American Academy of Allergy and Immunology (*Anonymous, 1991*). pBECs were derived by endobronchial brushing during fibre-optic bronchoscopy and cultured in hormonally supplemented BEGM containing 50 U/ml penicillin and 50 μg/ml streptomycin (*Wark et al., 2005*).

Virus stocks were grown in Vero cells as previously (*Caly et al., 2016*). A549 cells were grown for 12 hr before infection with either RSV A2 (denoted as RSV throughout), or eGFP-rRSV, a recombinant RSV expressing enhanced green fluorescent protein (eGFP) (*Webster Marketon et al., 2014*) in 2% FCS/HF-12K medium (multiplicity of infection (MOI) of 1–3). After 2 hr, cells were washed and media replaced; cells (or medium) at various times post infection (p.i.) were retained for analysis of cell-associated (or released) infectious virus (plaque forming units) and/or viral genomes (by quantitative PCR) as per (*Caly et al., 2016*).

## Immunofluorescence and confocal scanning laser microscopy (CLSM)

Mock- or RSV-infected A549 cells were stained with MitoTrackerRed CMXRos (M7512, ThermoFisher Scientific; 100 nM, 15 min) and then fixed, washed and stained using standard protocols (*Hu et al., 2013*). Primary antibodies used were: anti-RSV antibody (1:400, ab20745, abcam), anti-γ-tubulin antibody (1:300, 66320–1-Ig, Proteintech), or anti-α-tubulin (1:100, sc-5286, Santa Cruz), with dye-tagged secondary antibodies (anti-goat Alexa Fluor 488, 1:1000, A-11055, or anti-mouse Alexa Fluor 647, 1:1000, A-31571, ThermoFisher Scientific) as appropriate. F-actin was stained by Alexa Fluor 488 phalloidin (1:1000, A12379, ThermoFisher Scientific). In all analyses of stained fixed cells, nuclei were stained by DAPI (1:15,000 in PBS, 10236276001, Sigma). Following mounting onto glass slides with Biomeda Gel Mount (ProSciTech), imaging was conducted using a Leica TCS SP5 channel confocal and multiphoton microscope (63X objective, oil immersion). Images (512 × 512 pixels, 8- or 12-bit) were collected and viewed using the Leica Application Suite Advanced Fluorescence Lite Version: 2.8.0 build 7266 viewer software. Airyscan super-resolution imaging was performed using the Zeiss CLSM 800 with Airyscan detector; images (2448 × 2448 pixels, 16-bit) were viewed using the ZEN 2 (blue edition) software.

## Quantitative analysis of mitochondrial morphology and distribution

Mitochondrial morphologies were quantified as described (*Hu et al., 2013*). Briefly, fragmented, tubular or fibrillar mitochondrial morphologies were defined by width/length parameters of 1:1, 1:3 and 1:10 respectively. Quantification of each type of mitochondrial morphology in mock- or RSV-infected cells was assessed by counting 25–30 cells per condition on three independent occasions.

Quantitative analyses of mitochondrial organization and distribution were performed using custom scripts programmed in Python using *numpy* (*van der Walt et al., 2011*), *scipy*, *scikit-image* (*van der Walt et al., 2014*), *matplotlib* (*Hunter, 2007*) and *seaborn* (*Virtanen and Oliphant, 2016*; *Waskom et al., 2016*; see https://gitlab.erc.monash.edu.au/mmi/mito). To quantify mitochondrial perinuclear distribution, we measured the $R_{90\%}$ parameter, the radius of the circle required to enclose 90% of the MitoTrackerRed fluorescence relative to the centre of the nucleus (*van Bergeijk et al., 2015*); nuclei were segmented by applying a 2-pixel Gaussian filter and an Otsu threshold

(*Otsu, 1979*) to the DAPI channel (objects < 500 pixels were excluded). Mitochondria were then segmented by applying a 2-pixel Gaussian filter and a Li threshold (*Ch and Lee, 1993*) to the Mito-Tracker Red channel (objects < 10 pixels were excluded). Adjacent cells were split using a Watershed transform where nuclei centroids situated ≥10 pixels apart were used as markers. Infected cells analysed were those with mean anti-RSV-Alexa Fluor 488 fluoresence intensity 5–10 A. U., with manual inspection of the output. The $R_{90\%}$ was calculated by creating a Euclidean distance map using the nuclei centroids, which was masked using the segmented mitochondrial region to generate a map where the intensity of each pixel represents the distance of that pixel from the centre of the nucleus (*i.e.* the radius). A cumulative histogram was then constructed from pixel radial distances and $R_{90\%}$ calculated as the radius within which 90% of the mitochondrial pixels were contained.

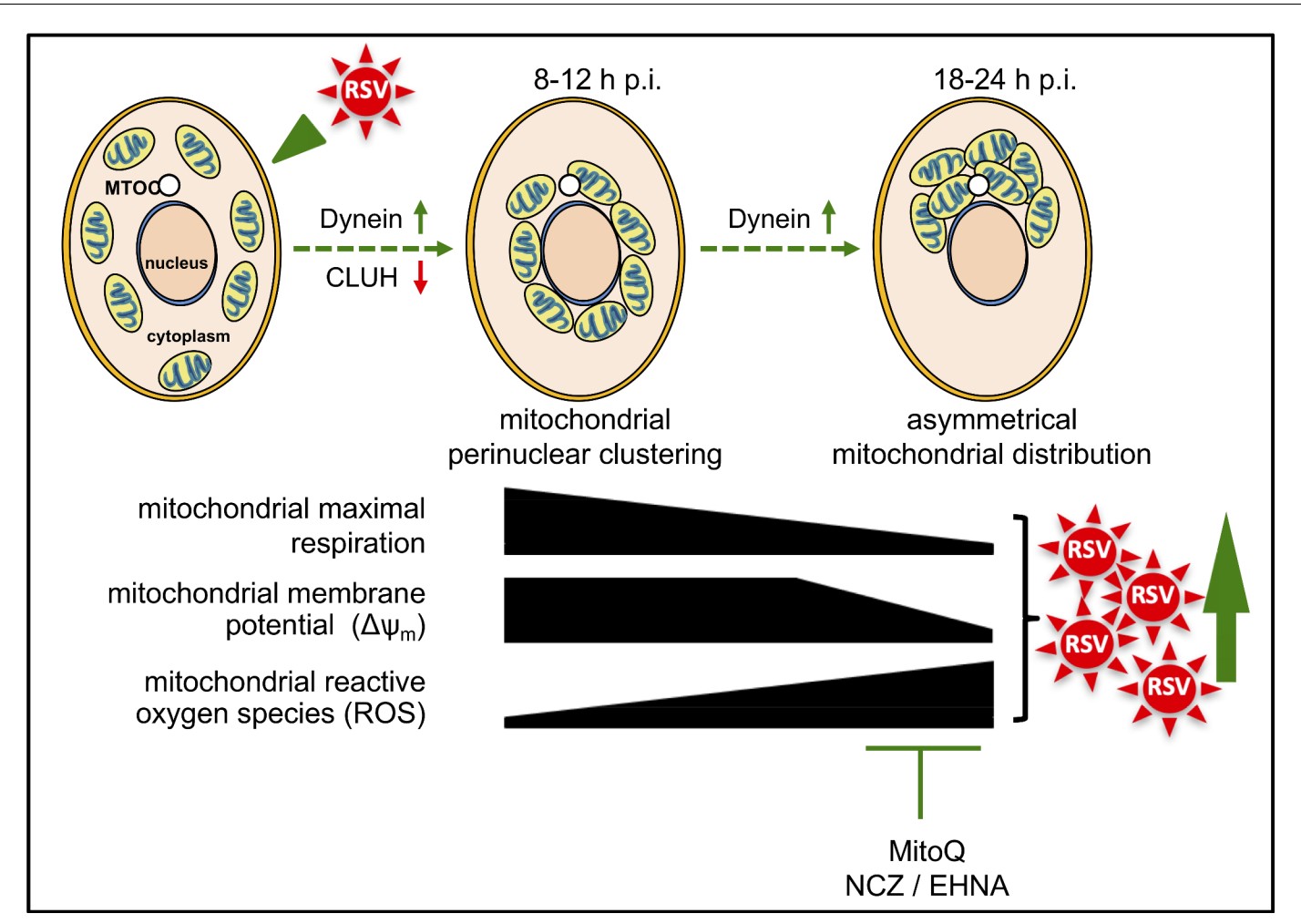

**Figure 10.** Schematic representation of the progressive host cell changes that favour RSV infection. RSV infection induces changes in mitochondrial organization with mitochondrial perinuclear clustering early in infection (8–12 h p.i.), followed by asymmetric distribution of mitochondria close to the MTOC later in infection (18–24 h p.i.); both phases of mitochondrial redistribution (top) are dependent on dynein components (inhibited by siRNAs directed at *DYNLT1* or *DYNC1H1*), with perinuclear clustering limited by CLUH (siRNA directed at *CLUH* increases perinuclear clustering, as well as mitochondrial ROS production and RSV virus production). Accompanying these changes, RSV infection inhibits host mitochondrial respiration, disrupts maintenance of mitochondrial membrane potential ($\Delta\psi_m$) and enhances mitochondrial reactive oxygen species (ROS) generation. These events favour RSV infection as indicated by fact that RSV infectious virus production is decreased by disrupting microtubule organization using nocodazole (NCZ), by inhibiting the dynein-motor with EHNA, or using the mitochondrially-targeted antioxidant MitoQ.
DOI: https://doi.org/10.7554/eLife.42448.023

To determine the angular distribution of mitochondrial pixels relative to the MTOC, the γ-tubulin channel was masked using the segmented mitochondrial regions as defined above. A peak detector, as implemented by the *scikit-image* peak_local_max function (*van der Walt et al., 2014*), was used to identify a single peak within the masked region, the coordinates of which were delineated as the MTOC. The orientation of each cell was then normalied such that the angle of the line between the nucleus centroid and the MTOC was equal to 0˚, that is the image was reoriented so that the MTOC was directly above the centroid of the nucleus. To analyse the angular distribution of mitochondria, the angle between the nucleus centroid and each pixel classified as mitochondria was calculated and plotted as a polar kernel density histogram. Finally, the total number of mitochondria classified pixels falling within 45˚ either side of the MTOC were extracted and presented as a proportion of total mitochondria classified pixels.

## siRNA interference

A549 cells were transfected using DharmaFECT transfection reagent (GE Dharmacon) in serum-free HF-12K medium with 50 nM siRNA (GE Dharmacon, SMART pool) targeting either cytoplasmic dynein 1 heavy chain 1 (*DYNC1H1*, #1778), dynein light chain Tctex type 1 (*DYNLT1*, #6993), kinesin light chain 1 (*KLC1*, #3831), clustered mitochondria homolog (*CLUH*, #23277), or scrambled control siRNA (GE Dharmacon). siRNA-transfected cells were maintained in a humidified 5% $CO_2$ atmosphere at 37˚C for 48 hr before infection.

## Immunoblot analysis

Cells were lysed and subjected to SDS-PAGE and immunoblot analysis as described (*Hu et al., 2013*). Specific proteins were detected using anti-DYNC1H1 (1:500, C-5, sc-514579, Santa Cruz), anti-DYNLT1 (1:1000, H-11, sc365567, Santa Cruz), anti-KLC1 (1:1000, L2, sc58776, Santa Cruz), anti-CLUH (1:1000, PA5-71324, ThermoFisher Scientific) or anti-α-tubulin (1:5000, B-7, sc5286, Santa Cruz) antibodies, together with anti-mouse HRP-coupled secondary antibody (1:10000, W4021, Promega).

## Assessment of mitochondrial bioenergetics and function

OCR (oxygen consumption rate) and ECAR (extracellular acidification rate) were monitored using the Seahorse XF96 Extracellular Flux Analyser (Seahorse Biosciences) (*Hu et al., 2013*). A549 cells were plated ($1 \times 10^4$ cells/well, 10% FCS/HF-12K) with or without RSV infection (MOI 1, 2% FCS/HF12K, 2 hr). Before the measurement, cells were washed twice with pre-warmed XF assay buffer (unbuffered DMEM supplemented with 25 mM glucose, 2 mM L-glutamine and 1 mM sodium pyruvate, pH 7.4) and then equilibrated in XF buffer (37˚C, 1 hr). Respiratory parameters for basal, ATP-linked, maximal uncoupled, spare and non-mitochondrial respiration were calculated from OCR in response to the sequential addition of 1 μM oligomycin (ATP synthase inhibitor), 1 μM FCCP (carbonyl cyanide p-trifluoromethoxyphenylhydrazone, proton ionophore), and a combination of 1 μM antimycin A (complex III inhibitor) and 1 μM rotenone (complex I inhibitor), respectively (*Hu et al., 2013*).

## Measurement of mitochondrial membrane potential ($\Delta\chi_m$) and ROS

$\Delta\psi_m$ was determined using $\Delta\psi_m$-sensitive fluorescent dyes. For imaging with tetramethylrhodamine ethyl ester (TMRE) (*Dejonghe et al., 2016*), A549 cells were mock- or eGFP-RSV-infected (MOI 1), with TMRE (ab113852, abcam; 50 nM, 15 min; Ex/Em: 561/565 ± 25 nm) with Hoechst (H3570, ThermoFisher Scientific; 5 μg ml$^{-1}$; Ex/Em: 405/470 nm) added for the last 5 min in the dark before imaging. For live cell imaging, A549 cells were mock- or RSV-infected (MOI 3) then incubated with tetraphenylethylene-phenyl-indolium salt (TPE-Ph-In) (2 μM, 30 min; Ex/Em: 488/680 ± 25 nm) (*Zhao et al., 2015*) in the dark before imaging over 16–18 hr. To minimize phototobleaching and phototoxicity to cells during imaging, measurements of $\Delta\psi_m$ and ROS were performed using a CLSM with 8 kHz resonant optical scanners (resonant scanning CLSM) for image resolution (512 × 512 pixels, 12-bit). TMRE fluorescence intensity was quantified from 15 to 20 cells in each treatment condition using Fiji (https://fiji.sc/).

Intracellular ROS production was visualised using dichlorodihydrofluorescein diacetate ($H_2$DCFDA/DCF, D399, ThermoFisher Scientific) (*Al-Mehdi et al., 2012*). A549 cells were mock- or RSV-infected (MOI 1), or treated with rotenone (0.5 μM, 30 min), and treated with or without MitoQ

(provided by Health Manufacturing, New Zealand [*Smith and Murphy, 2010*; *Maharjan et al., 2014*; 1 μM, 2 hr]), nocodazole (17 μM, 2 hr) or DMSO (vehicle) as a control. Cells were then incubated with Mitotracker Deep Red (M22426, ThermoFisher Scientific; 100 nM, 15 min; Ex/Em: 633/665 nm), with Hoechst and DCF (2.5 μM; Ex/Em: 496/517–527 nm) added at the last 5 min in the dark before imaging using resonant scanning CLSM over 8–18 hr.

Mitochondrial ROS was detected using the mitochondria-targeted ROS sensor, flavin-rhodamine redox sensor 2 (FRR2). A549 cells were mock- or RSV-infected (MOI 1) or treated with rotenone (0.5 μM, 30 min), MitoQ (1 μM, 2 hr) or DMSO (vehicle) as a control. Mitotracker Deep Red and FRR2 (2 μM, 15 min) with Hoechst (5 μg ml$^{-1}$) were added in the last 5 min before live cell imaging using resonant scanning CLSM at 8 or 18 hr. The ratiometric output of FRR2 (*Kaur et al., 2016*) ($I_{(Ex514)}/I_{(Ex488)}$; the ratio of the intensity of red emission [denoted as I] at 580 ± 20 nm upon excitation [Ex] at 514 nm *versus* 488 nm) serves a marker for mitochondrial ROS accumulation. Ratiometric $I_{(Ex514)}/I_{(Ex488)}$ images were generated by pixel-wise division of the 514 nm and 488 nm emission image channels using Fiji. For all samples, images were set to 32-bit float precision with a display range of min = 0.0 and max = 15.0 to facilitate comparison). To quantify the mitochondrial-localized ratio, a CellProfiler pipeline (http://cellprofiler.org/) was set up, whereby a pixel-wise image of $I_{(Ex514)}/I_{(Ex488)}$ was derived by pixel-wise division of the emission image channels acquired at 514 nm and 488 nm excitation, and stored as a 32-bit float image. Regions containing mitochondria were then segmented from the MitoTracker Deep Red channel by applying a five pixel Gaussian blur and an Otsu auto-threshold (*Otsu, 1979*), and then filtered to exclude all regions smaller than 1000 pixels. Segmented regions were then used to determine the mean ratiometric pixel value using the $I_{(Ex514)}/I_{(Ex488)}$ image above.

For FACS analysis, cells were trypsinised at different times, centrifuged, resuspended in FACS buffer (2% heat-inactivated FCS, 10 mM HEPES [(4-(2-hydroxyethyl)−1-piperazineethanesulfonic acid], 2 mM L-Glutamine, 2 mM EDTA solution) containing FRR2 (2 μM, 37℃, 15 min), and then analysed using a BD LSRII flow cytometer. Data analysis was performed using FlowJo software (Tree Star, Inc).

## Mouse model

All experiments were performed in accordance with The ACT Animal Welfare Act (1992) and the Australian Code of Practice for the Care and use of Animals for Scientific Purposes. The study protocol was approved by the Committee for Ethics in Animal Experimentation of the University of Canberra (project reference number CEAE 14–15). Groups of 5 BALB/c mice (6–8 weeks old) were infected intranasally with 2.5 × 10$^5$ pfu of recombinant RSV (rRSV) in 50 μl as described (*Foronjy et al., 2014*); control mice received 50 μl of viral diluent (mock). All mice were housed in cages covered with barrier filters and given water containing mitochondria-specific ROS scavenger MitoQ (500 μM) or fresh water *ad libitum* every day. Mice were monitored daily for signs of disease (lethargy, ruffled fur) and weight loss. On days 3, 4, 5 and 7, mice were sacrificed using cervical dislocation. One lung for each mouse was lysed in viral diluent with grinding beads using a TissueLyser II (Qiagen) for determination of viral genomes (by quantitative PCR) and infectious virus production (plaque forming units) per lung as per (*Caly et al., 2016*). The other lung from each mouse was fixed in formaldehyde, embedded in paraffin, sectioned and stained with haemotoxylin and eosin (H and E) (Imaging and cytometry Facility, John Curtin School of Medical Research, Australian National University, Canberra). Histological analysis of H and E-stained slides was used to determine bronchial and perivascular inflammation based on established quantification schema (*Ford et al., 2001*; *Mehra et al., 2012*). Briefly, the intensity of bronchial or perivascular inflammation was scored numerically for each view-field on a scale of 1 to 9. 0 denotes no inflammation; 1–3 scant cells but not forming a defined layer; 4–6, 1–3 layers of cells surrounding the airway epithelium or the vessel; and 7–9, four or more layers of cells surrounding the airway epithelium or the vessel. Blood from each mouse was also collected by heart puncture, serum extracted and used for ELISA to determine systemic inflammatory responses using RANTES as a marker, as per the manufacturer's specifications (RANTES, RayBiotech Inc).

## Cell viability assay

A cytotoxicity detection kit (LDH Release Assay, Roche Applied Science) was used to quantitatively assess cell death on the basis of the amount of LDH (lactate dehydrogenase) released into the medium upon plasma membrane damage. The LDH assay was carried out as previously (*Hu et al., 2013*) according to the manufacturer's instructions.

## Statistical analysis

All quantitative data in this study represent the mean value $\pm$ SEM for $n \geq 3$ (number of experiments). Significance levels were determined by ANOVA (GraphPad Prism 6).

## Acknowledgements

MH acknowledges the scholarship support by the University of Melbourne (Melbourne International Research Scholarship). The authors thank Michael N Teng (University of South Florida) for providing the eGFP-rRSV stock, Kristy Nichol (University of Newcastle, NSW, Australia) for providing immortalised and primary airway epithelial cells, and Health Manufacturing (New Zealand) for the generous donation of mitochondrial antioxidant MitoQ, mitoquinone mesylate. The authors also thank the Monash Micro Imaging Facility, (Monash University, Victoria, Australia), and acknowledge the financial support of the National Health and Medical Research Council Australia (Senior Principal Research Fellowship APP1002486/APP1103050 and Project grant APP1043511). The authors have no competing interests to declare.

## Additional information

### Funding

| Funder | Grant reference number | Author |
| --- | --- | --- |
| National Health and Medical Research Council | APP1002486 | David Andrew Jans |
| National Health and Medical Research Council | APP1043511 | David Andrew Jans |
| National Health and Medical Research Council | APP1103050 | David Andrew Jans |

The funders had no role in study design, data collection and interpretation, or the decision to submit the work for publication.

### Author contributions

MengJie Hu, Data curation, Formal analysis, Investigation, Methodology, Writing—original draft, Writing—review and editing; Keith E Schulze, Software, Formal analysis, Methodology; Reena Ghildyal, Investigation, Writing—review and editing; Darren C Henstridge, Investigation, Methodology; Jacek L Kolanowski, Yuning Hong, Methodology; Elizabeth J New, Peter AB Wark, Supervision; Alan C Hsu, Resources; Philip M Hansbro, Project administration; Marie A Bogoyevitch, Conceptualization, Supervision, Writing—review and editing; David A Jans, Conceptualization, Formal analysis, Supervision, Funding acquisition, Writing—original draft, Project administration, Writing—review and editing

### Author ORCIDs

MengJie Hu  https://orcid.org/0000-0002-7362-1452
Alan C Hsu  http://orcid.org/0000-0002-6640-0846
Marie A Bogoyevitch  https://orcid.org/0000-0001-9745-3716
David A Jans  https://orcid.org/0000-0001-5115-4745

## Ethics

Human subjects: Primary human bronchial epithelial cells (pBECs) were obtained from 4 healthy individuals who had no history of smoking or lung disease, had normal lung function, and gave written, informed consent to participate and have their data published, in accordance with the procedures in accordance with the procedures approved by the University of Newcastle Human Ethics Committee (project reference no. H-163-1205), in keeping with the guidelines of the National Institutes of Health, American Academy of Allergy and Immunology.

Animal experimentation: This study was performed in accordance with The ACT Animal Welfare Act (1992) and the Australian Code of Practice for the Care and use of Animals for Scientific Purposes. The study protocol was approved by the Committee for Ethics in Animal Experimentation of the University of Canberra (project reference number CEAE 14-15).

## Decision letter and Author response

Decision letter https://doi.org/10.7554/eLife.42448.028
Author response https://doi.org/10.7554/eLife.42448.029

# Additional files

## Supplementary files

• Transparent reporting form
DOI: https://doi.org/10.7554/eLife.42448.024

## Data availability

Data are being uploaded to Dryad (DOI: https://dx.doi.org/10.5061/dryad.2n3162c) Customised scripts for quantitative analyses of mitochondrial distribution and results are publicly available through https://gitlab.erc.monash.edu.au/mmi/mito (copy archived at https://github.com/elifesciences-publications/mito/).

The following dataset was generated:

| Author(s) | Year | Dataset title | Dataset URL | Database and Identifier |
|---|---|---|---|---|
| MengJie Hu, Keith E Schulze, Reena Ghildyal, Darren C Henstridge, Jacek L Kolanowski, Elizabeth J New, Yuning Hong, Alan C Hsu, Philip M Hansbro, Peter AB Wark, Marie A Bogoyevitch, David A Jans | 2019 | Data from: Respiratory Syncytial Virus co-opts host mitochondrial function to favour infectious virus production | https://dx.doi.org/10.5061/dryad.2n3162c | Dryad, 10.5061/dryad.2n3162c |

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
