## [Decision Letter]

Thank you for submitting your article "Respiratory Syncytial Virus co-opts host mitochondrial function to favour infectious virus production" for consideration by *eLife*. Your article has been reviewed by two peer reviewers, and the evaluation has been overseen by a Reviewing Editor and Anna Akhmanova as the Senior Editor. The following individual involved in the review of your submission has agreed to reveal his identity: Timothy Wai (Reviewer #2).

The reviewers have discussed the reviews with one another and the Reviewing Editor has drafted this decision to help you prepare a revised submission.

In this manuscript the link between mitochondrial reorganization and function and respiratory syncytial virus infection (RSV) is demonstrated. The authors show that RSV infection causes of redistribution of mitochondria that is microtubule-dependent. The redistribution then is associated with decreased mitochondrial respiration and increased reactive oxygen species (ROS) production. Furthermore, the authors show that microtubule disruption and ROS scavenging decreases RSV replication and that ROS scavenger treatment decreases RSV disease in mice.

The reviewers agree that the data are interesting and sound. They appreciate the novelty of the data concerning mitochondrial reorganization upon virus infection are novel. They also have critical points to be addressed during revisions.

Given that the disruption of microtubules may exert an indirect effect on virus assembly, no firm conclusions can be made without further experimentation. If the authors want to claim that mitochondrial redistribution affects viral replication, they should disrupt mitochondrial movement without altering microtubule formation. For example, preventing mitochondrial redistribution by targeting either CLUH, MIRO1 or 2 without disrupting microtubules would validate the conclusions.

Furthermore quantification of the data on mitochondrial morphology should be provided allowing more precise statements.

The authors should also check cell viability in the in vitro infection experiments to exclude the effects of cell death.

The findings should be critically verified and discussed in the light of the work done by the Casola group at UTMB. Specifically the experiments presented in the Figure 9 are an extension of the Casola group's antioxidant studies on RSV-induced disease, therefore cytokine/chemokine production, or other measures of antioxidant efficacy, should be done in a similar manner so that the results can be compared.

It would be great to provide evidence concerning mitochondrial redistribution in RSV-infected airway epithelial cells as this would add to our understanding by showing this effect occurs in vivo.

[Editors' note: further revisions were requested prior to acceptance, as described below.]

Thank you for resubmitting your work entitled "Respiratory Syncytial Virus co-opts host mitochondrial function to favour infectious virus production" for further consideration at *eLife*. Your revised article has been favorably evaluated by Anna Akhmanova as the Senior Editor, and a Reviewing Editor.

The manuscript has been improved but there are some remaining issues that need to be addressed before acceptance, as outlined below:

The reviewers agree that ROS generation mechanisms caused by mitochondrial reorganisation are beyond the scope of the study. However, a discussion of CLUH, ROS, and redistribution figure in the Discussion would further improve the message of the paper.

Furthermore, one of the reviewers notes that the authors have misinterpreted the first concern about microtubules and RSV replication. It is well known that disruption of microtubules or dynein motors decreases RSV replication by preventing the formation of filaments. Thus, the data presented in Figure 8 are not novel and do not directly support the study's hypothesis. However, the new CLUH siRNA data are novel and do support the hypothesis that mitochondrial reorganization is important for RSV replication. These new data should be presented as Figure 8 in place of the old one.

---

## [Author Response]

[…] The reviewers agree that the data are interesting and sound. They appreciate the novelty of the data concerning mitochondrial reorganization upon virus infection are novel. They also have critical points to be addressed during revisions.Given that the disruption of microtubules may exert an indirect effect on virus assembly, no firm conclusions can be made without further experimentation. If the authors want to claim that mitochondrial redistribution affects viral replication, they should disrupt mitochondrial movement without altering microtubule formation. For example, preventing mitochondrial redistribution by targeting either CLUH, MIRO1 or 2 without disrupting microtubules would validate the conclusions.

We thank the reviewers for this suggestion. In fact, in the original submission we had used two distinct approaches to impact mitochondrial movement without affecting microtubules directly; Figures 3A (second last row) and 3D (sixth and seventh rows) show results for a specific inhibitor targeting dynein ATPase activity, and siRNA knockdown of a dynein adaptor protein or cytoplasmic dynein heavy chain component, respectively. In all of these, treatment restores the even distribution of mitochondria in RSV-infected cells without affecting the filamentous microtubule structure, as indicated clearly by the α-tubulin staining. We now highlight in the text more clearly that these treatments do *not* disrupt microtubules, but clearly impact the effects of RSV on infected cell mitochondria (subsections “RSV-induced mitochondrial redistribution is microtubule- and dynein-dependent” and “RSV infection inhibits host mitochondrial respiration dependent on dynein/microtubule integrity”). To build on the reviewers’ specific suggestion, however, we have now added completely new experimental data (five panel Figure 8—figure supplement 6), where we use siRNA to knockdown the CLUH protein, as suggested. The results are most exciting, showing that CLUH knockdown induces mitochondrial perinuclear clustering (even in the absence of RSV infection) (Figure 8—figure supplement 6B, C), resulting in elevated mitochondrial ROS (Figure 8—figure supplement 6D, E), and most importantly, resulting in increased RSV virus production (Figure 8—figure supplement 6F). Together with Figure 3A, D, this demonstrates that mitochondrial redistribution (and not microtubule disruption per se) is key to mitochondrial ROS production, which in turn facilitates RSV virus production (subsection “Mitochondrial redistribution/elevated mitochondrial ROS generation is critical for RSV virus production”, last paragraph). We thank the reviewers again for encouraging us to perform these experiments.

Furthermore quantification of the data on mitochondrial morphology should be provided allowing more precise statements.

We thank the reviewers for encouraging us to do more analysis; we now include a new panel in Figure 1 (Figure 1B) showing the quantitative analysis. This supports the conclusions from Figure 1A, with significantly altered mitochondrial morphology evident in virus-infected cells compared to mock-infected cells. We have also amended our description of the results, to make them more precise (subsection “RSV infection drives mitochondrial perinuclear clustering and redistribution of 43 mitochondria towards the microtubule organizing centre (MTOC)”, first paragraph), as requested.

The authors should also check cell viability in the in vitro infection experiments to exclude the effects of cell death.

We thank the reviewers for this suggestion; we now include a new figure (Figure 8—figure supplement 3) that documents the minimal effects on cell viability of our various siRNA and drug treatments in mock- and RSV-infected cells. The results confirm that the observed reduction in viral replication is not a result of reduced cell viability (subsection “Mitochondrial redistribution/elevated mitochondrial ROS generation is critical for RSV virus production”). We thank the reviewers for encouraging us to present this analysis.

The findings should be critically verified and discussed in the light of the work done by the Casola group at UTMB. Specifically the experiments presented in the Figure 9 are an extension of the Casola group's antioxidant studies on RSV-induced disease, therefore cytokine/chemokine production, or other measures of antioxidant efficacy, should be done in a similar manner so that the results can be compared.

The reviewers’ comments highlight the clear parallels between our studies, and those of the Casola group. Although our experiments examine distinct aspects of inflammation (airway, perivascular, and systemic; Figure 9) as opposed to inflammation of the bronchoalveolar lavage (Casola group), both studies show reduced levels of chemokine (RANTES) as well as lung inflammation following antioxidant treatment in RSV-infected mice compared to the control group (Figure 9G; compare to Figure 6, PMID: 17008643). Importantly, both studies support the idea that antioxidant treatment ameliorates RSV-induced pathogenesis. To add emphasis to this point, we now include detailed discussion of how our results relate to and confirm those of the Casola group (Discussion, second paragraph), as well as adding additional references (including from the Casola group). These changes have made the relevance of the manuscript to the present literature much clearer, and help feature the work of the Casola group, as appropriate.

It would be great to provide evidence concerning mitochondrial redistribution in RSV-infected airway epithelial cells as this would add to our understanding by showing this effect occurs in vivo.

We thank the reviewers for this suggestion. We now include new figures, namely Figures 1—figure supplements 2 and 3, documenting the key observations of RSV-induced mitochondrial perinuclear clustering in human airway progenitor-like basal cells, including quantitative analysis, and Figure 8—figure supplement 1, documenting the fact that the mitochondrial ROS scavenger MitoQ protects against RSV infection in human primary airway epithelial cells (4 lines). The results confirm the physiological relevance of our results in the A549 system, providing important insights into RSV’s impact on mitochondrial distribution of human airway epithelia (subsections “RSV infection drives mitochondrial perinuclear clustering and redistribution of mitochondria towards the microtubule organizing centre (MTOC)” and “RSV-induced mitochondrial redistribution is microtubule- and dynein-dependent”; Discussion, second paragraph).

[Editors' note: further revisions were requested prior to acceptance, as described below.]

The manuscript has been improved but there are some remaining issues that need to be addressed before acceptance, as outlined below:The reviewers agree that ROS generation mechanisms caused by mitochondrial reorganisation are beyond the scope of the study. However, a discussion of CLUH, ROS, and redistribution figure in the Discussion would further improve the message of the paper.

We have expanded discussion of CLUH, ROS and mitochondrial redistribution in the Discussion section, as requested. Importantly, we have included CLUH in Figure 10 (absent in the previous revision) and in the legend. We thank the reviewers for this suggestion.

Furthermore, one of the reviewers notes that the authors have misinterpreted the first concern about microtubules and RSV replication. It is well known that disruption of microtubules or dynein motors decreases RSV replication by preventing the formation of filaments. Thus, the data presented in Figure 8 are not novel and do not directly support the study's hypothesis. However, the new CLUH siRNA data are novel and do support the hypothesis that mitochondrial reorganization is important for RSV replication. These new data should be presented as Figure 8 in place of the old one.

To satisfy the reviewers, we have swapped the supplementary data on CLUH (Figure 8—figure supplement 1 in the previous submission) with the data in Figure 8. We have translocated our novel data from the previous Figure 8C into Figure 7E, as well as reshuffle the order of appearance in the manuscript of several of the supplementary figures to enhance the flow. Importantly, we now refer appropriately to previous work on RSV filament formation to preface the section on microtubules and RSV replication (new reference Vanover et al., 2017). We thank the reviewer for these suggestions.